# Cryo-EM structures of the human Elongator complex at work

Nour-el-Hana Abbassi [1,2,9], Marcin Jaciuk [1,9], David Scherf [3], Pauline Böhnert [3], Alexander Rau [4], Alexander Hammermeister [1], Michał Rawski [1,5], Paulina Indyka [1,5], Grzegorz Wazny[5,6], Andrzej Chramiec-Głąbik[1], Dominika Dobosz[1], Bozena Skupien-Rabian [1], Urszula Jankowska [1], Juri Rappsilber [4,7], Raffael Schaffrath [3] ✉, Ting-Yu Lin [1,8] ✉ & Sebastian Glatt [1] ✉

tRNA modifications affect ribosomal elongation speed and co-translational folding dynamics. The Elongator complex is responsible for introducing 5-carboxymethyl at wobble uridine bases ($cm^5U_{34}$) in eukaryotic tRNAs. However, the structure and function of human Elongator remain poorly understood. In this study, we present a series of cryo-EM structures of human ELP123 in complex with tRNA and cofactors at four different stages of the reaction. The structures at resolutions of up to 2.9 Å together with complementary functional analyses reveal the molecular mechanism of the modification reaction. Our results show that tRNA binding exposes a universally conserved uridine at position 33 ($U_{33}$), which triggers acetyl-CoA hydrolysis. We identify a series of conserved residues that are crucial for the radical-based acetylation of $U_{34}$ and profile the molecular effects of patient-derived mutations. Together, we provide the high-resolution view of human Elongator and reveal its detailed mechanism of action.

The four basic nucleotide building blocks of RNA can undergo numerous post-transcriptional modifications at various positions in the backbone and base[1]. Over 170 unique modifications have been identified in RNA families from all domains of life[2] and ~100 of them are found in tRNAs[3,4]. Modifications impact tRNA folding and regulate their function in various biological processes[5,6]. For instance, numerous modifications in the anticodon stem loop (ASL) are required for accurate ribosomal translation and regulate co-translational folding dynamics of the resultant nascent polypeptide chains[7,8]. Position 34, also known as the "wobble position", is a modification hotspot and carries a variety of modifications in different tRNAs[7,9]. In eukaryotes, wobble

uridines ($U_{34}$) are modified to 5-carbamoylmethyluridine ($ncm^5U$), 5-methoxycarbonylmethyluridine ($mcm^5U$), 5-methoxycarbonyl-1-hydroxymethyluridine ($mchm^5U$), carbamoylmethyl-2′-O-methyluridine ($ncm^5Um$), 5-methoxycarbonylmethyl-2′-O-methyluridine ($mcm^5Um$) or 5-methoxycarbonylmethyl-2-thiouridine ($mcm^5s^2U$). These bulky chemical groups facilitate non-canonical base pairing between the tRNA anticodon and the mRNA codon in the A site of the ribosome. Hence, $U_{34}$ modifications strongly contribute to accurate decoding[10], facilitating an optimal speed of mRNA translation[11] and supporting proteome homeostasis[12–14]. Loss of these modifications hampers cellular function, increases sensitivity to stress conditions[11], or causes embryonic lethality

[1]Małopolska Centre of Biotechnology (MCB), Jagiellonian University, Krakow, Poland. [2]Postgraduate School of Molecular Medicine, Medical University of Warsaw, Warsaw, Poland. [3]Institute for Biology, Department for Microbiology, University of Kassel, Kassel, Germany. [4]Bioanalytics, Institute of Biotechnology, Technical University of Berlin, Berlin, Germany. [5]SOLARIS National Synchrotron Radiation Centre, Jagiellonian University, Krakow, Poland. [6]Doctoral School of Exact and Natural Sciences, Jagiellonian University, Krakow, Poland. [7]Wellcome Centre for Cell Biology, University of Edinburgh, Edinburgh, UK. [8]Present address: Department of Biosciences, Durham University, Durham, UK. [9]These authors contributed equally: Nour-el-Hana Abbassi, Marcin Jaciuk. ✉e-mail: schaffrath@uni-kassel.de; ting-yu.lin@durham.ac.uk; sebastian.glatt@uj.edu.pl

in mouse models[15]. Changes in $U_{34}$ modification levels have also been linked to cancer, neurodegenerative diseases, and neuronal dysfunction[16].

The multi-subunit protein complex Elongator[17] is responsible for the conversion of $U_{34}$ to $cm^5U_{34}$ in eukaryotes, the priming modification for other subsequent $U_{34}$-modifications (e.g., $ncm^5U_{34}$, $mcm^5U_{34}$). The highly conserved complex is composed of two copies of each of its six individual protein subunits, Elp1-Elp6, which form two discrete subcomplexes known as Elp123 and Elp456. Elp1 is the main scaffolding subunit and dimerizes via its C-terminus, forming an "arch" structure with an Elp123 lobe on either side, each harboring one enzymatically active Elp3 subunit[18]. Elp3 contains two functional domains, a radical *S*-adenosylmethionine (rSAM) and a lysine acetyltransferase (KAT) domain[19]. The modification reaction requires *S*-adenosylmethionine (SAM) and acetyl-CoA as cofactors. tRNA binding to Elp123 triggers SAM cleavage and acetyl-CoA hydrolysis that provide the required reactive groups for $cm^5U_{34}$ formation[20]. The associated Elp456 subcomplex forms a hexameric ring, which interacts asymmetrically with only one lobe of Elp123[18,21,22]. Elp456 is thought to use its ATPase activity to release the modified tRNA from the Elongator complex[18,23].

Several structures of yeast and mouse Elp123 with bound tRNA have recently been determined. However, the mechanistic details of how active site residues trigger the Elp123-mediated modification reaction and how cofactors are accommodated in the catalytic cavity remain elusive. In this study, we reconstructed high-resolution single-particle cryo-electron microscopy (cryo-EM) structures of recombinant human ELP123 in complex with tRNA, SAM, and acetyl-CoA representing various stages of the reaction. We examined the functional role of conserved active site residues using biochemical and biophysical approaches and confirmed the findings using in vivo yeast reporter assays. Unexpectedly, we discovered that $U_{33}$, a universally conserved uridine present in almost all tRNAs, plays an essential role for Elongator's activity in vitro and in vivo. We used crosslinking mass spectrometry to analyze the assembly of the two human Elongator subcomplexes and examined pathogenic patient-derived mutations to characterize their impact on Elongator function. In summary, we provide a detailed structural and mechanistic analysis of the human Elongator, disclose several unprecedented details of its tRNA modification activity, and provide high-resolution information about a relevant drug target for future therapeutic approaches.

## Results

### Cryo-EM structures of human ELP123 subcomplex with bound tRNA and cofactors

While our recent work has demonstrated a functional and structural conservation of the Elongator complex from yeast (*Saccharomyces cerevisiae*, *Sc*) to mouse (*Mus musculus*, *Mm*)[23], numerous variants of human (*Homo sapiens*, *Hs*) Elongator associated with pathology emphasize the need for a comprehensive characterization of the structure and tRNA modification activity of human ELP123. ELP123 can be produced in insect cells[24,25], and so we first co-purified the recombinant ELP123 subcomplex with high purity, showing the expected 1:1:1 stoichiometry of the three subunits (Fig. 1a).

In humans, 13 different tRNA species that harbor a U at position 34 are targeted by the Elongator complex. Hence, we profiled the ability of ELP123 to select modifiable tRNAs while rejecting non-modifiable ones at the binding stage. Using microscale thermophoresis (MST), we measured the affinity of Elongator towards various in vitro-transcribed tRNAs, including $tRNA^{Lys}_{UUU}$, $tRNA^{Glu}_{UUC}$, $tRNA^{Arg}_{UCU}$, and $tRNA^{Ser}_{UGA}$ and the non-modifiable $tRNA^{Gln}_{CUG}$. The calculated $K_d$ values for $tRNA^{Ser}_{UGA}$, which has a longer variable loop, and $tRNA^{Gln}_{CUG}$, which is not modified by Elongator, are the weakest ($137.7 \pm 10.5$ nM and $117.7 \pm 13.8$ nM, respectively). The calculated affinity of ELP123 towards in vitro-transcribed human $tRNA^{Gln}_{UUG}$ is the highest ($37.6 \pm 5.3$ nM) among the

tested human tRNAs (Fig. 1b). We also confirmed that purified ELP123 does not bind any of the histone- and tubulin-derived peptides previously suspected to undergo Elongator-dependent lysine acetylation[26] (Fig. 1b). The presence of acetyl-CoA, *S*-ethyl-CoA (ECA, a non-hydrolysable acetyl-CoA analog), or coenzyme A (CoA) does not enhance or decrease tRNA binding. Of note, the presence of desulpho-CoA (DCA), which mimics the acetyl-CoA hydrolysis product, lowers the affinity between the complex and $tRNA^{Gln}_{UUG}$ ($182.9 \pm 36.3$ nM; Fig. 1c).

Next, we conducted a structural characterization of the ELP123–tRNA–acetyl-CoA complex using single-particle cryo-EM. We vitrified the freshly purified complex after incubation with ligands (tRNA, acetyl-CoA, and SAM), screened the prepared grids, and collected a complete dataset. The obtained movies were motion- as well as CTF-corrected and suitable particle sets were identified during iterative rounds of 2D/3D classification steps (Fig. S1). After 3D reconstruction and refinement, the obtained density resembles a 2-lobed architecture, known from other eukaryotic Elp123 complexes[23,27]. However, while one ELP123 lobe is fully resolved the second lobe is only partially visible. To improve the quality of the final reconstruction, we masked out the less well-defined lobe and locally refined the remaining region[23]. The final map has an overall resolution of 2.87 Å ($FSC_{0.143}$; Fig. 1d and Table 1), with some regions reaching a local resolution of 2.6 Å (Fig. S1f). Since the less-resolved lobe shows weak density for a second bound tRNA molecule, we used focused 3D subclassification to verify that human ELP123 can indeed bind two tRNA molecules in both lobes at the same time (Fig. 1d).

We used individual Alphafold2[28] predictions of ELP1, ELP2, and ELP3 as a starting point for rounds of flexible fitting[29] and manual building steps to converge into an atomic model of the human ELP123 complex. The refined structure has a characteristic "moth-like" architecture[18,22], resembling previously reported structures of the yeast and mouse counterparts. We confirm that the lobe structure is facilitated by two WD40 domains of ELP1 and two WD40 domains of ELP2, which clamp the catalytic ELP3 subunit from opposite sides. The tRNA is bound by ELP3 and the C-terminal tetratricopeptide domain (TPR) of ELP1[23,27]. However, the TPR of ELP1 displays lower local resolution, indicating either a weaker interaction of this region with the T-arm or high intrinsic structural flexibility even in the presence of bound tRNA. In addition, we prepared samples of ELP123 without tRNA and solved its structure at 3.58 Å resolution. The reconstructed map only resolves the WD40 domains of ELP1 as well as ELP2 and ELP3 (Fig. S2a–f). The lack of interpretable density for the TPR of ELP1 again indicates its high flexibility in the absence of tRNA[23] (Fig. S2g).

In summary, we determined high-resolution cryo-EM structures of the human ELP123 subcomplex with and without human $tRNA^{Gln}_{UUG}$, confirming that Elp123 subcomplexes from yeast, mouse, and human are structurally highly conserved and bind tRNAs in a similar fashion.

### The acetyl-CoA loop in ELP3 coordinates acetyl-CoA in its KAT domain

In our structures, the KAT domain of human ELP3, which binds the acetyl-CoA ligand, is similar to known Elp3 structures[18,19,30] and to the crystal structure of the GCN5-type acetyltransferases, which bind and utilize acetyl-CoA to acetylate lysine residues of histones[31]. Our previously reported structures of a bacterial Elp3 homolog demonstrated the structural mechanism by which it coordinates DCA in its active site. However, there has been no reported experimental evidence showing how eukaryotic Elongator complex or archaeal Elp3 proteins bind acetyl-CoA (or its analogs). In our ELP123–tRNA–acetyl-CoA reconstruction, we identified a well-resolved ligand density that unambiguously fits a complete acetyl-CoA molecule, confirming that the native ligand is bound in a similar fashion as acetyl-CoA in GCN5 and DCA in bacterial *Dmc*Elp3 (Fig. S3a). The acetyl group of acetyl-CoA reaches deep into the catalytic cleft of ELP3. Acetyl-CoA is predominantly bound through its phosphate groups, which are coordinated by a

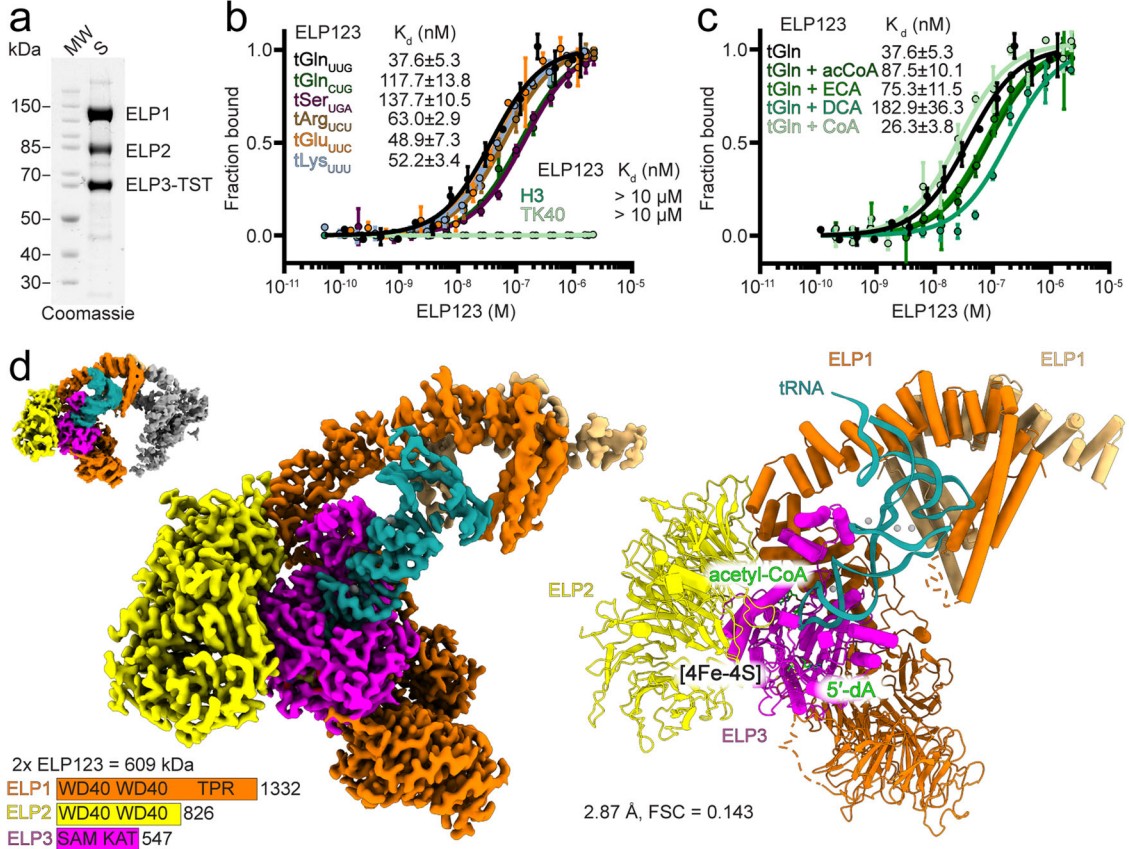

**Fig. 1 | Cryo-EM structure of human ELP123–tRNA$^{Gln}_{UUG}$–acetyl-CoA complex.**
**a** SDS-PAGE analysis of purified ELP123 subcomplex. Predicted molecular weights (MW): ELP1 (148 kDa), ELP2 (93 kDa), ELP3 with Twin-Strep-Tag (TST; 65 kDa). **b** MST measurements with calculated dissociation constant ($K_d$) values for ELP123 bound to various tRNAs. $n = 3$ (independent experiments). Data are presented as mean values ± SEM. **c** MST measurements with $K_d$ values for ELP123 bound to tRNA$^{Gln}_{UUG}$ in the absence or presence of ligands (500 μM), including acetyl-CoA

(acCoA), desulpho-Coenzym A (DCA) and S-Ethyl-Coenzyme A (ECA). $n = 3$ (independent experiments). Data are presented as mean values ± SEM. **d** Cryo-EM density map (left) and atomic model (right) of ELP123 subcomplex bound to tRNA and acetyl-CoA (ELP1, orange; ELP1 second lobe, light orange; ELP2, yellow; ELP3, magenta and tRNA$^{Gln}_{UUG}$, deep teal) at a resolution of 2.87 Å. The two-lobe subclass density map is shown in the top left corner. Source data are provided as a Source Data file.

Gly-Gly motif (Gly496 and Gly498), and several residues located in a loop of ELP3 (aa477-497), hereafter called the "acetyl-CoA loop". Notably, the acetyl-CoA loop is not visible in the tRNA-free ELP123 structure (Fig. 2a), suggesting that it is highly mobile and does not block the acetyl-CoA pocket the way the analogous loop does in bacterial *Dmc*Elp3. Next, we determined individual structures of human ELP123–tRNA$^{Gln}_{UUG}$ in the presence of ECA (3.35 Å resolution) or DCA (4.25 Å resolution) to assess the complex architecture throughout the acetyl-CoA hydrolysis reaction (Fig. 2a and Figs. S3b–g and S4). These structures show that the overall architecture of ELP3 does not change as the acetyl-CoA reaction proceeds, with the acetyl-CoA loop continuing to coordinate the acetyl-CoA (or analog) substrate. However, for both ECA- and DCA-bound samples, we found a larger number of particles in which the TPR of ELP1 loses contact with the T-arm (Figs. S3g and S4e). This effect is more pronounced for ECA-containing samples, where the resulting structure completely lacks the TPR region. In summary, we show how acetyl-CoA is coordinated in the active site of human ELP123, confirming the importance of specific active site residues and following their conformational changes throughout different stages of the hydrolysis reaction. Furthermore, the presence of tRNA and acetyl-CoA stabilizes the conformation of the TPR domain of ELP1, while acetyl-CoA analogs induce the flexibility of this part of the subcomplex. These observations suggest that the human ELP123 complex is more dynamic than its yeast counterpart and that the complex gets ready to release the tRNA after acetyl-CoA hydrolysis.

Although the overall structure of ELP3 and the acetyl-CoA loop remain unchanged during acetyl-CoA hydrolysis, we noticed subtle conformational changes of certain residues in the acetyl-CoA binding pocket (Fig. 2a). To gain further insight into the tRNA-induced activation of the acetyl-CoA hydrolysis reaction, we performed a structure-guided mutational analysis of active site residues in human ELP3. We focused on Glu474 and Tyr530 in the vicinity of the acetyl group that is freed during the reaction, as well as Lys164 and His476, which lie in close proximity to the acetyl-CoA long chain, and also Tyr529, the only residue whose side chain interacts with the acetyl-CoA diphosphate group. Of note, we also confirmed that histone- and tubulin-derived peptides do not trigger the acetyl-CoA hydrolysis activity of ELP123 (Fig. S5a). We produced and purified human ELP123 variants from insect cells carrying specific substitutions of these active site residues, namely ELP3$_{K164A}$, ELP3$_{E474A}$, ELP3$_{H476A}$, and ELP3$_{Y529A/Y530A}$ (Fig. S5b, c). Since tRNA binding is a prerequisite of acetyl-CoA hydrolysis by ELP3[20,30], we first measured the tRNA binding parameters of the ELP123 variants. We show that these mutants retain similar tRNA binding affinities to wild-type ELP123 (37.6 ± 5.3 nM; Fig. 2b) and only the ELP3$_{Y529A/Y530A}$ mutant has a slightly reduced affinity (94.7 ± 5.2 nM). In contrast, the tRNA-induced acetyl-CoA hydrolysis activity is reduced 3–4 fold for all tested mutants compared to wild-type, confirming the direct involvement of the tested residues in the hydrolysis reaction (Fig. 2c).

We observed the presence of an iron-sulfur cluster in all obtained structures, coordinated by three highly conserved cysteines (Cys99,

**Table 1 | Cryo-EM data collection, refinement, and validation statistics**

|  | ELP123–tRNA–acetyl-CoA (PDB ID 8PTX) (EMD-17924) | ELP123 (PDB ID 8PTY) (EMD-17925) | ELP123–tRNA–ECA (PDB ID 8PTZ) (EMD-17926) | ELP123–tRNA–DCA (PDB ID 8PU0) (EMD-17927) |
|---|---|---|---|---|
| **Data collection and processing** | | | | |
| Magnification | 105,000× | 96,000× | 96,000× | 105,000× |
| Voltage (kV) | 300 | 300 | 300 | 300 |
| Electron exposure (e–/Å$^2$) | 40.84 | 40.00 | 40.00 | 41.22 |
| Defocus range (μm) | −0.9 to −2.1 | −0.9 to −2.7 | −0.9 to −2.7 | −0.9 to −2.1 |
| Pixel size (Å) | 0.86 | 0.86 | 0.85 | 0.86 |
| Symmetry imposed | C1 | C1 | C1 | C1 |
| Initial particle images (no.) | 1,250,013 | 264,351 | 130,531 | 699,211 |
| Final particle images (no.) | 215,353 | 171,951 | 31,251 | 115,026 |
| Map resolution (Å) | 2.87 | 3.58 | 3.35 | 4.25 |
| FSC threshold | 0.143 | 0.143 | 0.143 | 0.143 |
| Map resolution range (Å) | 1.85 to >10 | 3.30 to 5.50 | 1.82 to >10 | 3.79 to >10 |
| **Refinement** | | | | |
| Initial model used (PDB code) | | | | |
| Model resolution (Å) | 1.8 | 3.2 | 2.1 | 3.0 |
| FSC threshold | 0.143 | 0.143 | 0.143 | 0.143 |
| Map sharpening B factor (Å$^2$) | DeepEMhancer | 120.408 | DeepEMhancer | DeepEMhancer |
| Model composition | | | | |
| Non-hydrogen atoms | 24,378 | 14,189 | 17,194 | 22,416 |
| Protein residues | 2852 | 1786 | 1965 | 2612 |
| Nucleotides | 75 | – | 75 | 71 |
| Ligands | 7 | 2 | 5 | 3 |
| B factors (Å$^2$) | | | | |
| Protein | 176.79 | 177.76 | 136.67 | 260.36 |
| Nucleotide | 243.32 | – | 251.00 | 243.61 |
| Ligand | 118.09 | 182.75 | 121.26 | 258.17 |
| R.m.s. deviations | | | | |
| Bond lengths (Å) | 0.005 | 0.005 | 0.004 | 0.005 |
| Bond angles (°) | 0.982 | 0.973 | 0.993 | 0.958 |
| Validation | | | | |
| MolProbity score | 1.70 | 2.09 | 2.05 | 2.33 |
| Clashscore | 6.26 | 11.81 | 10.62 | 23.61 |
| Poor rotamers (%) | 0.04 | 0.13 | 0.23 | 0.61 |
| Ramachandran plot | | | | |
| Favored (%) | 94.88 | 91.36 | 91.39 | 92.64 |
| Allowed (%) | 5.12 | 8.64 | 8.61 | 7.36 |
| Disallowed (%) | 0.00 | 0.00 | 0.00 | 0.00 |

Cys109, and Cys112) (Fig. S5d). Although SAM was added to all tRNA-containing samples, we did not see intact SAM molecules in the obtained structures but rather found possible SAM cleavage products: 5-deoxyadenosyl (5′-dA) was bound via Arg367, and a bulky density resembling methionine was bound via the iron-sulfur cluster (Fig. S5d). 5′-dA and methionine are the products of the reductive SAM cleavage reaction triggered by tRNA binding[32]. Together, our structural analyses suggest that in human ELP123–SAM cleavage proceeds constitutively in the rSAM domain and independent of acetyl-CoA hydrolysis by the KAT domain.

## ELP3 and ELP1 mediated recognition of tRNAs
The high quality of the ELP123–tRNA–acetyl-CoA map allowed us to precisely trace almost the entire tRNA molecule de novo. We find well-defined densities for most tRNA nucleotides, with a slightly weaker coverage in the tRNA elbow region, as well as the acceptor stem that points away from the complex and does not make any direct contacts with ELP123 (Fig. 3a). By comparing the structure of ELP123-bound tRNA$^{Gln}_{UUG}$ with the crystal structure of free tRNA$^{Phe}$ (PDB 1EHZ), we infer that the ASL is significantly distorted upon binding to ELP123 in the catalytic cleft (Supplementary Movie S1).

The N-terminus of ELP3 is relatively poorly conserved[30], and it has remained unclear how relevant this region is for tRNA binding. From our structures, we see that the N-terminal extension of ELP3 predominantly interacts with the D-arm of the tRNA substrate. Unlike previous lower-resolution structures, we are able to interpret the specific conformation of the N-terminus of ELP3 and build a complete atomic model de novo. The N-terminus of ELP3 contains a three-helix bundle that positions several basic residues to sense the shape of the tRNA and contact the tRNA backbone. While the main body of the

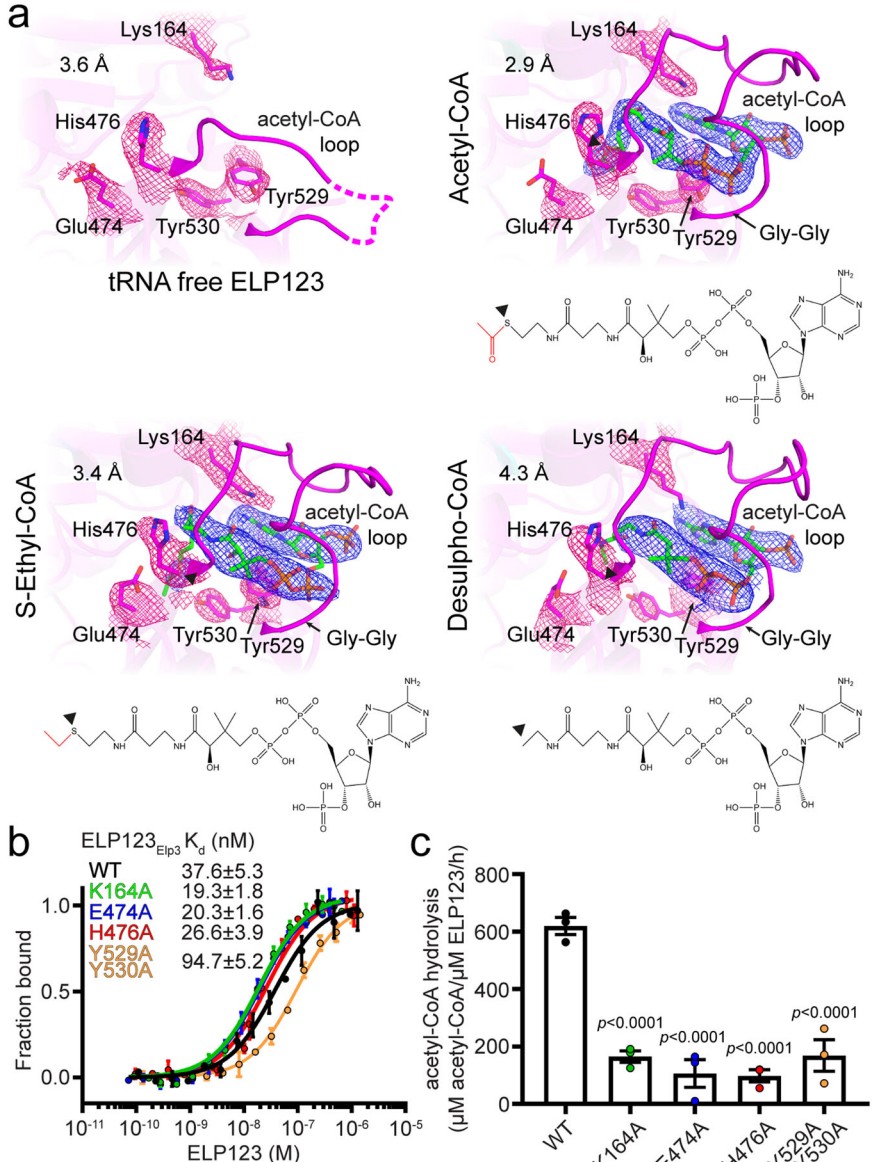

**Fig. 2 | Structure and biochemical characterizations of conserved acetyl-CoA binding and hydrolysis. a** Close-up view of acetyl-CoA loop in the tRNA-free state of ELP123 compared to ELP123 bound to tRNA in the presence of acetyl-CoA (2.87 Å), ECA (3.35 Å) or DCA (4.25 Å). In all close-ups, residues interacting with acetyl-CoA and their respective densities are highlighted (pink mesh) while the densities of the ligands are shown in blue. **b** MST measurements with calculated $K_d$ values for ELP123 and ELP3 mutants bound to tRNA$^{Gln}_{UUG}$. $n = 3$ (independent experiments). Data are presented as mean values ± SEM. **c** Acetyl-CoA hydrolysis rates of ELP123 and ELP3 variants in the presence of tRNA$^{Gln}_{UUG}$, $n = 3$ (independent experiments). Statistical analysis: one-way ANOVA with Dunnett's multiple comparisons test. Statistically significant differences are indicated. Data are presented as mean values ± SEM. Source data are provided as a Source Data file.

N-terminus (aa6-75) binds the central part of the tRNA, aa76-85 are in close proximity to the ASL bound in the active site of ELP3 with direct tRNA contacts made by residues Lys42, Thr43, Gln54, Arg56, Lys79, Arg82, and Ser85. We speculate that these interactions might be responsible for the local deformation and stabilization of the distorted conformation of the ASL. We also compared rigid body-fitted models of the N-termini from mouse, yeast, archaeal, and bacterial Elp3 proteins to show that their topology is indeed similar (Fig. S6a). In addition to the interaction with the N-terminus, several conserved arginine residues in the catalytic cleft (Arg151, Arg242, Arg361, Arg364, Arg367, Arg384, and Arg402) contact the phosphate backbone of the ASL (Fig. 3a).

To validate the contribution of individual regions for tRNA recognition and the activity of ELP123, we generated structure-guided mutants, including a truncation of the N-terminus (ELP3$_{\Delta1-85}$) and alanine substitutions of the most centrally positioned basic residues in the catalytic cleft of ELP3 (ELP3$_{R361A}$ and ELP3$_{R364A}$). We used the purified ELP123 variants to measure their binding affinity by MST. All of them show lower affinities (2–4 fold lower than ELP123$_{WT}$) towards tRNA$^{Gln}_{UUG}$ and lead to a complete loss of tRNA-induced acetyl-CoA hydrolysis, indicating that not only tRNA binding but also specific distortions of the ASL are required for inducing the acetyl-CoA hydrolysis activity (Fig. 3b).

The tRNA elbow region, which is formed by T- and D-arms, is known to be contacted by a basic loop in the C-terminus of yeast Elp1[33]. In previous yeast and mouse cryo-EM structures (EMD-4574, EMD-15623, and EMD-15625), Elp1 contacts tRNA through disordered and flexible loop regions. However, human ELP1 interacts with the tRNA mainly through helices that are formed by aa1103–1150 and aa1215–1246, exposing specific basic residues that contact the RNA backbone. Moreover, in the unfiltered map, we can find a weak density in close proximity to the tRNA D-arm and variable loop (nucleotides

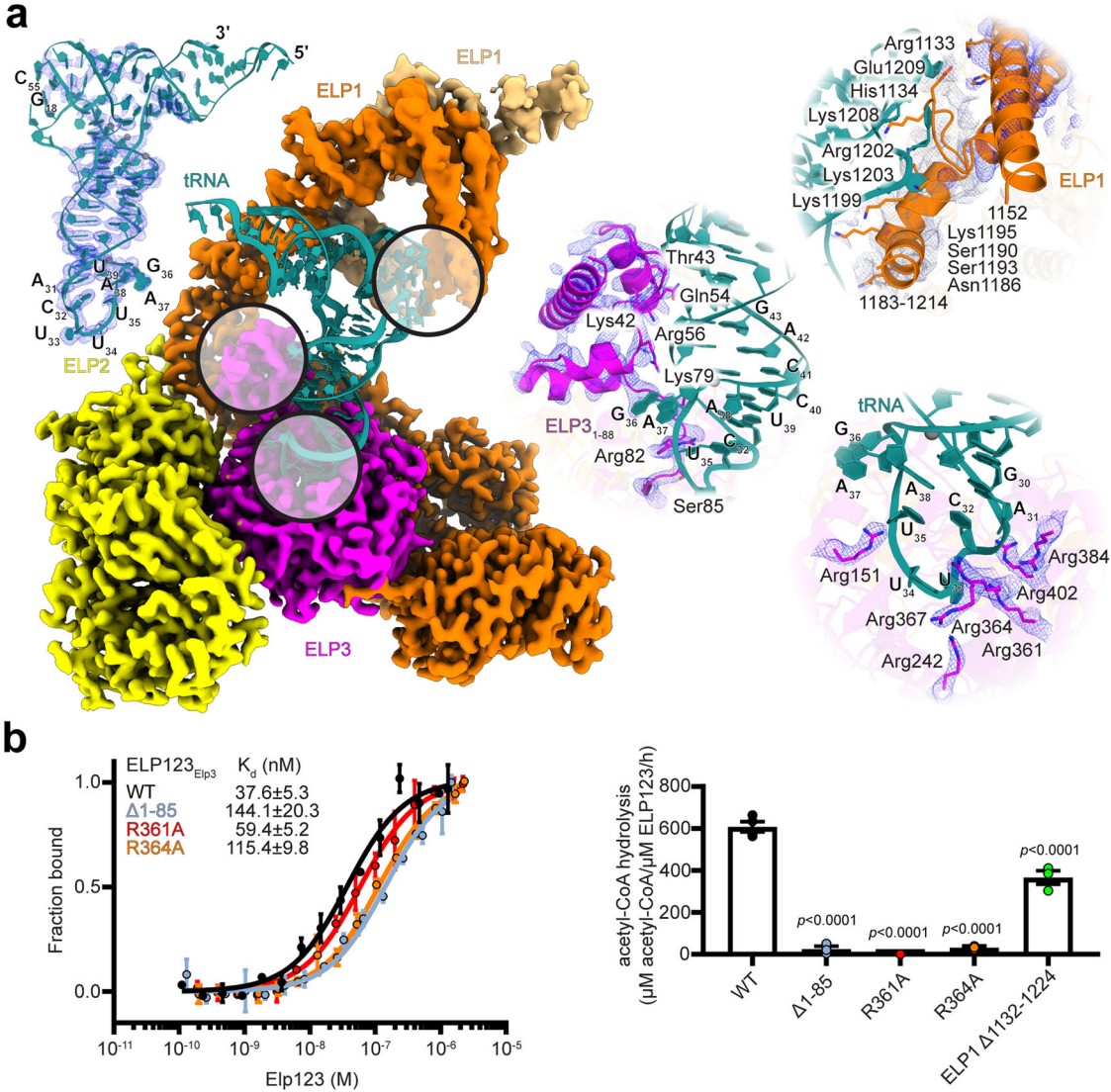

**Fig. 3 | Inter-subunit coordination of tRNA binding. a** Close-up views of tRNA recognition by ELP1 and ELP3. The density of the full tRNA is shown while the three major contacts are displayed. The close-up views of contacting residues with tRNA interaction are shown on the right. Sharpened map of ELP3 in blue and unfiltered map of ELP1 in gray. **b** Left: MST measurements with calculated $K_d$ values for ELP123 and ELP3 mutants bound to tRNA$^{Gln}_{UUG}$. $n = 3$ (independent experiments). Right: Acetyl-CoA hydrolysis rates of ELP123 and ELP1 and ELP3 mutants in the presence of tRNA$^{Gln}_{UUG}$, $n = 3$ (independent experiments). Statistical analysis: one-way ANOVA with Dunnett's multiple comparisons test. Statistically significant differences are indicated. Data are presented as mean values ± SEM. Source data are provided as a Source Data file.

20–22 and 42–45, respectively), that fits yet another predicted helix (aa1183–1214, Fig. 3a). Surprisingly, when we deleted the tRNA interacting region from the ELP1 C-terminus (Δ1132–1224), tRNA binding was not affected[23], but acetyl-CoA hydrolysis dropped by approximately 50% in comparison to wild-type ELP123 (Fig. 3b). In summary, we conclude that the main interaction between tRNA and ELP123 occurs via ELP3 and the poorly conserved N-terminus of ELP3 is crucial for the high-affinity. Moreover, we speculate that the dynamic interaction between the T-arm of tRNA and the ELP1 C-terminus might be required for the initial recognition steps of the modification reaction.

**The invariable U$_{33}$ triggers acetyl-CoA hydrolysis**
We have shown that the ASL is significantly distorted upon binding to ELP123 (Supplementary Movie 1)[23,27], with an almost identical architecture across the acetyl-CoA-, DCA-, and ECA-bound structures (Fig. S6b). We infer from this that the ASL does not undergo major conformational changes during the initial steps of the acetyl-CoA hydrolysis reaction. The deformation of the ASL upon binding not only

positions the target tRNA residue U$_{34}$ in close proximity to the iron-sulfur cluster[23,27] but also places U$_{33}$ close to the KAT domain and acetyl-CoA (Fig. S6c). His476 in the KAT domain faces U$_{33}$, but no contacts between the KAT domain and U$_{34}$ can be found (Fig. S6c). As the 33rd position of tRNAs is predominantly occupied by uridines[34] and almost never modified[35], we hypothesized that U$_{33}$ may be a key factor for ELP123 to identify tRNA substrates and trigger acetyl-CoA hydrolysis. To test this hypothesis, we generated several variants of tRNA$^{Gln}_{UUG}$ by replacing uridines at the 33rd or 34th position in the ASL with C (U$_{33}$C$_{34}$UG, C$_{33}$U$_{34}$UG, and C$_{33}$C$_{34}$UG). We confirmed that all variants of tRNA$^{Gln}$ are bound by ELP123 with comparable affinities (Fig. 4b). Strikingly, replacing U$_{33}$ with C abolishes tRNA-induced acetyl-CoA hydrolysis, while the substitution of U$_{34}$ does not severely affect acetyl-CoA hydrolysis activity (Fig. 4c).

Given that the presence of U$_{33}$ is essential for stimulating acetyl-CoA hydrolysis of ELP123 in vitro, we sought to investigate if U$_{33}$ is indeed required for Elongator activity in vivo. We used yeast harboring the suppressor tRNA *SUP4* (tRNA$^{SUP4}$): a mutated tRNA$^{Tyr}$ carrying a U$_{34}$

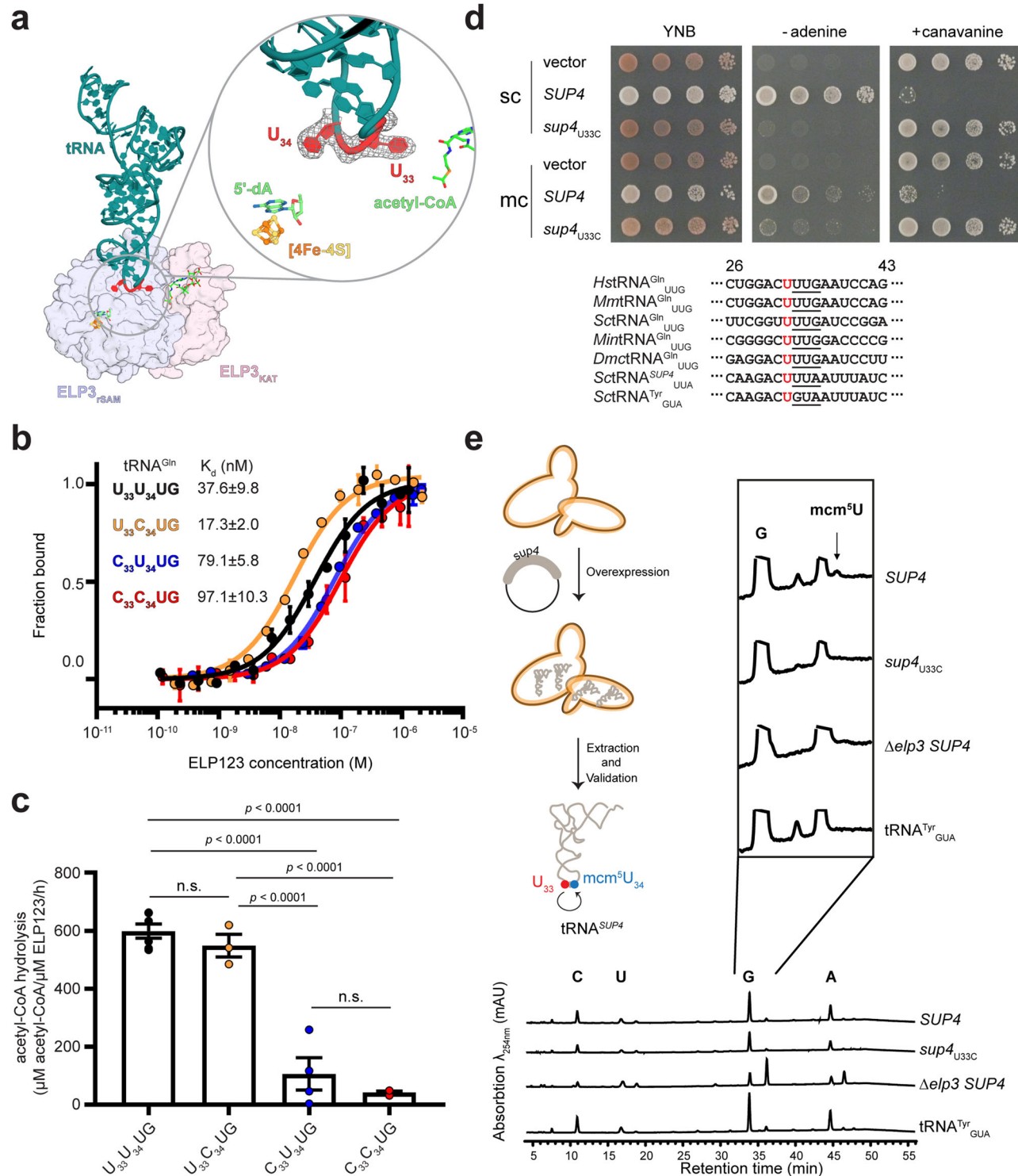

**Fig. 4 | Biochemical and functional characterizations of U₃₃ and its role for U₃₄ modification.** **a** Cartoon representations of tRNA bound to ELP3 and close-up view of the ASL where $U_{33}$ and $U_{34}$ are highlighted. The rSAM and KAT domains of ELP3 are color-coded. **b** MST measurements with calculated $K_d$ values for ELP123 bound to variants of tRNA$^{Gln}$. $n = 3$ (independent experiments). Data are presented as mean values ± SEM. **c** Acetyl-CoA hydrolysis rates of ELP123 in the presence of variants of tRNA$^{Gln}$. $n = 3$ (independent experiments). Statistical analysis: one-way ANOVA with Tukey's multiple comparisons test. Statistically significant differences are indicated (n.s.: not significant). Data are presented as mean values ± SEM. **d** Top: Phenotypic

analyses of various yeast strains without or with overexpression of tRNA$^{SUP4}$ and tRNA$^{sup4}$_{U33C}$ using UAA *ochre* nonsense readthrough assays (sc: single copy; mc: multicopy). Elongator-modified *SUP4* produces canavanine sensitivity and adenine prototrophy. Bottom: Sequence alignments of the ASL region in tRNA$^{Gln}$ from various species as well as yeast tRNA$^{Tyr}$ and tRNA$^{SUP4}$. The invariant $U_{33}$ is highlighted while the anticodon is underlined. **e** HPLC analysis of modified tRNA nucleosides. tRNA was extracted from various yeast strains expressing tRNA$^{Tyr}_{GUA}$, tRNA$^{SUP4}$, and tRNA$^{sup4}$_{U33C}$. The Δ*elp3* deletion strain is used as a control. The peak of mcm⁵U is indicated by the arrow. Source data are provided as a Source Data file.

instead of $G_{34}$ in the native $tRNA^{Tyr}$, allowing it to be modified by Elongator. The resulting $mcm^5U_{34}$ modification is crucial for UAA nonsense (*ochre*) suppression activity, which has previously been applied for monitoring the status of $U_{34}$ modification in vivo[36]. To examine the impact of $U_{33}$ on the modification levels of $U_{34}$ in vivo, we overexpressed either WT $tRNA^{SUP4}$ or a mutant carrying $C_{33}$ instead of $U_{33}$ ($sup4_{U33C}$) and monitored the efficiency of *ochre* readthrough using two reporter genes: *ade2-1* and *can1-100*. The yeast strain carrying WT $tRNA^{SUP4}$ formed white colonies and the cells were sensitive to canavanine treatment, indicating the presence of $mcm^5$ on $U_{34}$ of *SUP4* and *ochre* readthrough in the *ade2-1* and *can1-100* reporters, respectively (Fig. 4d). In contrast, cells expressing $sup4_{U33C}$ tRNA were red and resistant to canavanine due to premature translation termination of the *ade2-1* and *can1-100 ochre* reporters, respectively (Fig. 4d). Next, we extracted and purified $tRNA^{SUP4}$ from the different yeast strains and analyzed the modification patterns by high-pressure liquid chromatography (HPLC) (Fig. 4e and Fig. S7). We identified a characteristic elution peak for $mcm^5U$ in WT $tRNA^{SUP4}$, which disappears when either yeast Elp3 is deleted or cells express mutated $sup4_{U33C}$ tRNA. In summary, we provide strong evidence that Elongator uses the universally conserved $U_{33}$ nucleotide as a crucial determinant for Elongator-mediated $mcm^5U_{34}$ formation in vitro and in vivo.

## Conserved residues in ELP3 transfer the acetyl group across the catalytic cleft

Our structural and biochemical results show that the presence of $U_{33}$ is essential for acetyl-CoA hydrolysis, which in turn generates an activated acetyl group for the subsequent carboxymethylation of $U_{34}$. However, the acetyl group is generated ~25 Å away from the attachment site on the fifth carbon of $U_{34}$. As the position of $U_{34}$ does not seem to change during the reaction, we next asked how the acetyl group could be transported across the catalytic cleft. In our high-resolution cryo-EM structures, we noticed the presence of several conserved lysine and tyrosine residues (Lys280, Tyr363, Lys316, and Tyr318) lining up between acetyl-CoA and SAM just beneath $U_{33}$ and $U_{34}$ (Fig. 5a). Since lysine residues can undergo canonical acetylation, we hypothesized that the acetyl group could be transiently deposited on Lys280 proximal to the acetyl-CoA binding site and relayed along this lysine-rich stretch to eventually be installed on the fifth carbon of $U_{34}$ with assistance from the SAM-derived radical 5′-dA.

To investigate the proposed mechanism, we dissected the acetylation profiles of purified ELP123 proteins using mass spectrometry. We were able to detect acetylation of several lysine and tyrosine residues along the proposed relay in wild-type ELP3, including Lys280, Lys316, and Tyr318, in addition to several other acetylation sites distant from the active site (Fig. S8). While we did not detect acetylation at Tyr363, this could be due to the small size of the peptide fragment generated by digestion. Biochemical characterization of $ELP3_{K280A}$ and $ELP3_{Y363A}$ mutants revealed that they retain tRNA binding affinities (Fig. 5b) but exhibit significant defects in acetyl-CoA hydrolysis activity in vitro (Fig. 5c). As we did not observe any densities for acetylated lysine residues in our cryo-EM maps, we speculate that the acetyl transfer might be highly transient. Next, we employed γ-toxin tRNase assays to detect the presence of modified tRNAs harboring $mcm^5s^2U_{34}$[37] in yeast strains carrying mutations of these highly conserved residues in the catalytic cleft. Based on resistance to growth inhibition by the Elongator-dependent tRNase toxin in vivo (Fig. 5d) and loss of tRNA cleavage in vitro (Fig. S9), mutating any of the identified residues (Tyr136, Lys289, Lys325, Tyr327, Tyr489) in yeast Elp3 abolishes $U_{34}$ modification in vivo.

Decreased acetyl-CoA hydrolysis rates were also observed for mutations in the rSAM domain, including $ELP3_{C109S/C112S}$ (iron-sulfur cluster coordination), $ELP3_{R367A}$ (SAM binding) and $ELP3_{Y318A}$ (radical transfer). As those residues are not directly involved in acetyl-CoA hydrolysis or tRNA interaction, we conclude that these substitutions

likely impact the overall assembly of the catalytic site and affect ASL coordination. Their functional role is further evidenced by phenotypes in vivo, showing resistance to growth inhibition and tRNA cleavage by γ-toxin (Fig. 5d and Fig. S9). Of note, there are several other highly conserved residues in the catalytic pocket and substituting these with alanine ($ELP3_{Y136A}$, $ELP3_{E230A}$, $ELP3_{E253A}$, $ELP3_{H293A}$, $ELP3_{Y372A}$, $ELP3_{H485A}$, $ELP3_{H487A}$, and $ELP3_{Y489A}$) also diminishes in vivo $U_{34}$ modification (Fig. S9). This notion indicates possible alternative routes for the acetyl transfer, pointing to a complex network within the catalytic cleft that is ultimately necessary for the modification reaction.

## Conserved ELP123 and ELP456 interaction

Human ELP123 has been shown to interact with ELP456 in vitro[24], and we therefore attempted to reconstitute the full human Elongator assembly for cryo-EM analysis. Despite extensive efforts, stabilization of the samples by mild chemical crosslinking, and different preparation methods, we were not able to identify any particles resembling the fully assembled complex in the micrographs. To overcome this difficulty, we explored the use of UV-induced chemical crosslinking mass spectrometry (crosslinking MS)[23] to study human Elongator. First, we validated the methodology with purified reconstituted mouse Elongator complex (Fig. S10). The majority of detected self and heteromeric crosslinks indeed agree with the atomic model of mouse Elongator (EMD-15626)[23]. Assuming that the architecture is conserved between mouse and human, we generated a reference model of human Elongator by flexible fitting the structure of human ELP123 (Fig. 1d) and the model of human ELP456[38] into a 9 Å low-pass filtered map of mouse Elongator[23]. Next, we analyzed the reconstituted human Elongator complex by crosslinking MS. In agreement with the model, most of the observed crosslinks are found between residues that are in the acceptable distance range of 30 Å. However, we did find an increased number of violated crosslinks between ELP123 and ELP456, indicating increased flexibility and dynamics between the two human subcomplexes. Although this observation could explain why we did not obtain a reasonable cryo-EM map of the human complex, it remains unclear whether this is caused by the reconstitution protocol or the inherent dynamics of the human subcomplexes. To sum up, we show that yeast, mouse, and human Elongator bear the same overall architecture and that the interaction interfaces between the subcomplexes are conserved. Moreover, the crosslinking MS analyses confirm our previous observation that the purified reconstituted human Elongator complex is structurally highly flexible.

## Characterization of pathogenic ELP3 variants

Several ELP3 missense mutations are described in genome variant databases, including ClinVar[39], International Cancer Genome Consortium[40], and The Cancer Genome Atlas (https://www.cancer.gov/ccg/research/genome-sequencing/tcga). ELP3 variants have also been linked to neurodegenerative diseases, such as amyotrophic lateral sclerosis (ALS)[41]. To understand how these mutations affect Elongator activity, we analyzed ALS-related (R454K, R473K) and cancer-related (I298S, R242K, R402T, and D443N) variants of human ELP3. The presented human ELP123 structure allows to directly map these residues without the need for homology modeling (Fig. 6a and Fig. S11). We found that Arg242 and Arg402 are located in the catalytic cleft and seem to be responsible for the binding of the ASL and coordination of $U_{33}$. Ile298 is positioned at the base of the rSAM domain, Arg454, and Arg473 are located in the interaction interface between KAT and rSAM domains, and Asp443 is found in the KAT domain close to the TPR of ELP1.

Among these six mutants, we were able to produce and purify R242K and R402T, whereas R454K, R473K, I298S, and D443N either yield insoluble proteins or ELP123 subcomplexes with incorrect stoichiometry. Difficulties in obtaining ELP123 subcomplexes of these variants indicate the importance of these residues for the stability of

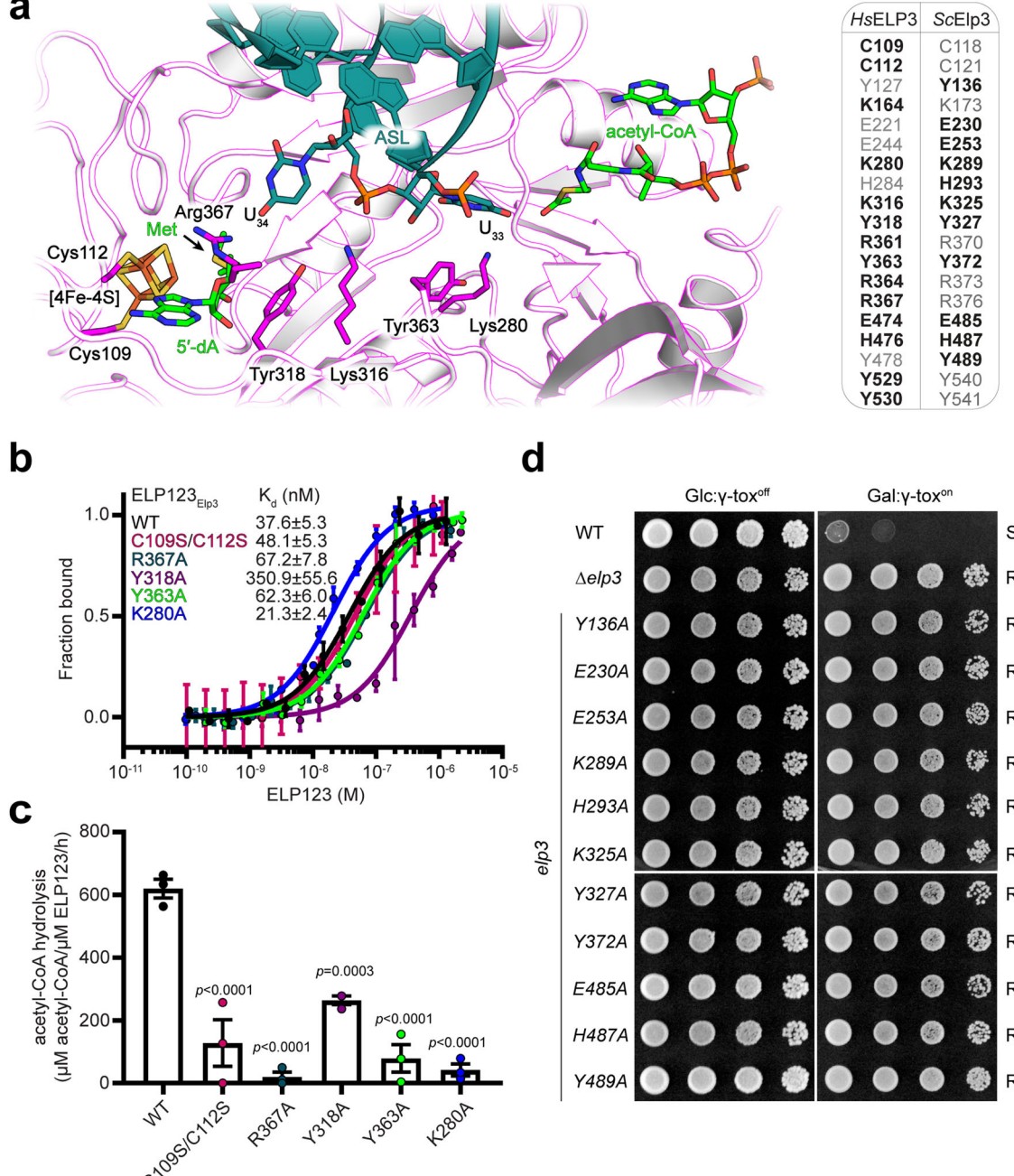

**Fig. 5 | Identification of a cluster of residues responsible for acetyl group transfer across ELP3 domains. a** Close-up view of the catalytic site of ELP3 with residues for Fe-S and SAM binding as well as acetyl-CoA transfer highlighted. The conserved residues in the catalytic site of ELP3, including human and yeast, are listed in the inlet. The tested residues in this study are highlighted in bold. **b** MST measurements with calculated $K_d$ values for ELP123 bound to tRNA$^{Gln}_{UUG}$. $n = 3$ (independent experiments). Data are presented as mean values ± SEM. **c** Acetyl-CoA hydrolysis rates of ELP123 and ELP3 mutants in the presence of tRNA$^{Gln}_{UUG}$. $n = 3$ (independent experiments). Statistical analysis: one-way ANOVA with Dunnett's multiple comparisons test. Statistically significant differences are indicated. Data are presented as mean values ± SEM. **d** Phenotype of yeast strains with Elp3 variants in response to galactose (Gal) induced γ-toxin expression. Sensitivity ('S') and resistance ('R') traits to growth inhibition by the Elongator-dependent tRNase toxin are appropriately labeled. Cell growth under glucose (Glc) conditions repressing toxin expression served as negative control. Source data are provided as a Source Data file.

the ELP3 subunit. We characterized tRNA binding parameters and tRNA-induced acetyl-CoA hydrolysis rates of R242K and R402T. Both mutants retain similar affinities to tRNA, but display significantly decreased acetyl-CoA hydrolysis rates (Fig. 6b, c). In summary, we used structural information about the human Elongator complex to directly map patient-derived mutations and employed biochemical assays to show that the mutations affect the stability or activity of the complex. Our high-resolution structure therefore paves the way to analyze and predict the relevance of Elongator variants in clinical diagnostics.

## Discussion

The function of the Elongator complex was originally linked to the acetylation of lysine residues in histones, tubulin, and other proteins[42,43]. However, overwhelming evidence has shown that Elongator rather acts as a tRNA modifying enzyme that acetylates wobble uridines[13]. All previously associated deletion phenotypes in model systems can be rescued by the overexpression of certain tRNA species[44]. A recent study discovered a moonlighting function of Elongator during cytokinesis when translation is turned off, in which

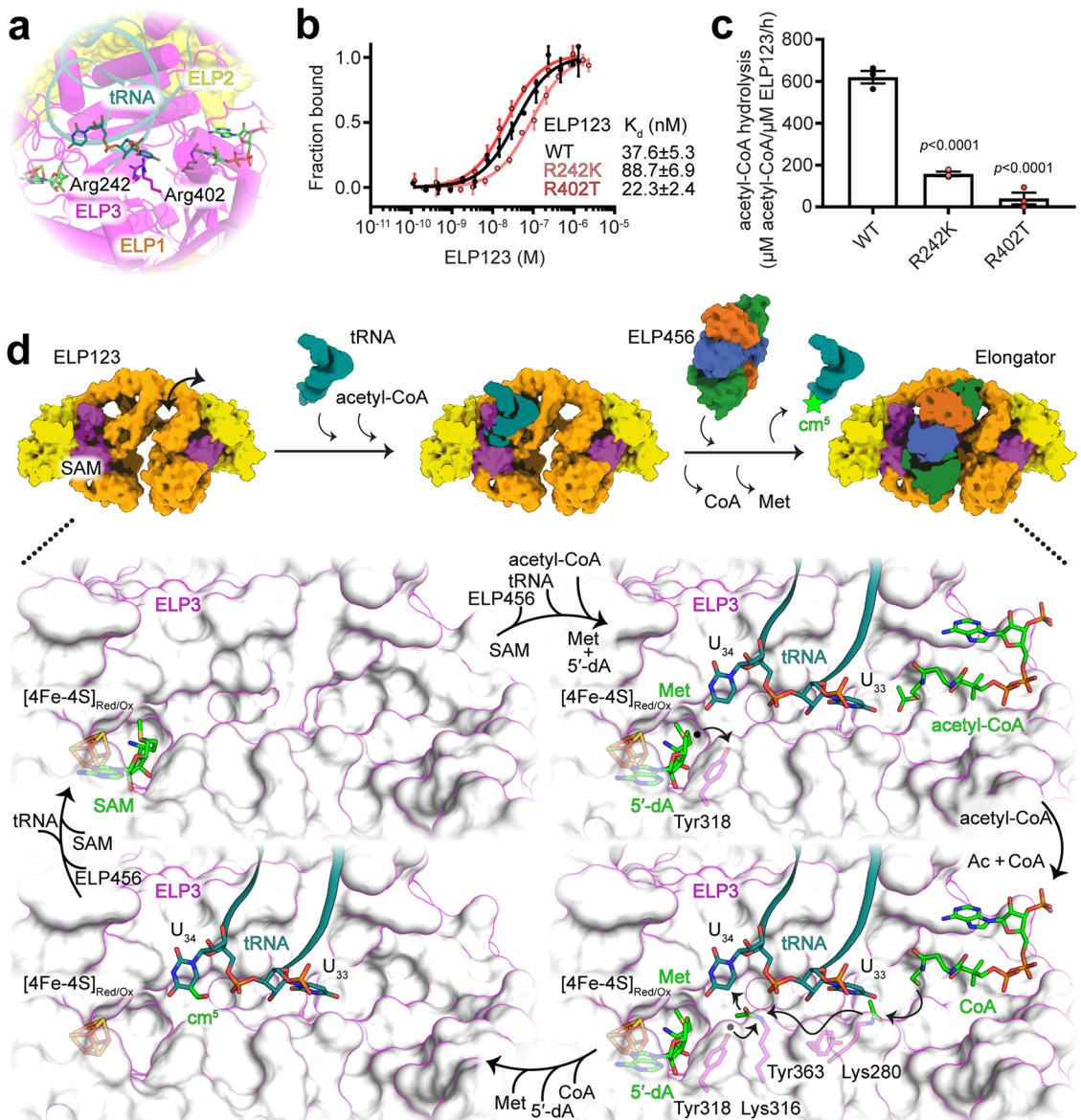

**Fig. 6 | Overview of human Elongator complex-mediated reaction. a** Mapping of the analyzed clinically relevant ELP3 variants. **b** MST measurements with calculated $K_d$ values for clinically relevant ELP3 variants bound to tRNA$^{Gln}_{UUG}$. $n = 3$ (independent experiments). Data are presented as mean values ± SEM. **c** Acetyl-CoA hydrolysis rates of clinical-relevant ELP3 variants in the presence of tRNA$^{Gln}_{UUG}$, $n = 3$ (independent experiments). Statistical analysis: one-way ANOVA with Dunnett's multiple comparisons test. Statistically significant differences are indicated. Data are presented as mean values ± SEM. **d** Proposed mechanism of cm$^5$ addition

by human Elongator. While S-Adenosylmethionine (SAM) is in place, the tRNA binds to ELP123 and triggers the recruitment of acetyl-CoA. SAM is cleaved into 5′-dA and methionine (MET) and the radical reaches Tyr318. From the other side, U$_{33}$ triggers the acetyl-CoA hydrolysis and the acetyl group (Ac) travels through the channel with the help of Lys280, Tyr363 to reach Lys316. From there the Tyr318 radical and the acetyl group are close enough for addition on U$_{34}$. ELP456 facilitates displacement of the modified tRNA by binding to the ELP123. Source data are provided as a Source Data file.

Elongator seems to affect microtubule dynamics to regulate cytoskeleton polarity and asymmetric signaling in sensory organ precursor cells of flies[45]. Strikingly, this function of Elongator appears to be independent of its acetyl-CoA hydrolysis and tRNA modification activity. The mechanistic details of how the complex binds to microtubules and which types of microtubules are affected[46] need to be thoroughly investigated in the future. Despite our excitement that Elongator fulfills an important cellular function outside of tRNA-mediated regulation of protein synthesis, our presented work focuses on understanding the canonical function of Elongator in tRNA modification. The structure of the human Elongator complex and its catalytic regulation have remained elusive and all structural as well as biochemical work has been limited to Elp proteins from model organisms such as yeast, fly, and mouse. Given the clinical relevance of

patient-derived mutations in Elongator subunits, it is of prime importance to understand the mechanistic details of the human Elongator complex.

Our current work addresses this knowledge gap by presenting a series of high-resolution cryo-EM structures of the human ELP123 subcomplex trapped in various early stages of the reaction. We present the initial state (ELP123−SAM), substrate-bound state (ELP123−tRNA−acetyl-CoA/ECA), and the post-modification reaction (ELP123−tRNA−DCA), which enable us to propose a sequential order of action (Fig. 6d). First, the initial state of ELP123 exhibits high structural mobility when it probes for potential tRNA substrates. Second, tRNA binding occurs via the ASL in the active site of ELP3, while the N-terminus of ELP3 binds to the D-loop and stabilizes the open ASL conformation. Third, the C-terminal region of ELP1 assists tRNA

binding through interactions with the elbow region of the tRNA. Binding into the catalytic cleft distorts the ASL and promotes the flipping-out of both, the modifiable $U_{34}$ base and the universally conserved $U_{33}$ base. This conformation triggers acetyl-CoA hydrolysis and the release of the acetyl group from acetyl-CoA. The acetyl group is relayed via conserved residues from the KAT domain to Lys316 in the rSAM domain. The 5′-dA radical formed by the iron-sulfur cluster-mediated SAM cleavage is transferred towards the acetyl group by Tyr318 to form an acetyl radical. Finally, the activated acetyl group is attached to the 5$^{th}$ carbon of $U_{34}$ to form cm$^5$U$_{34}$. After the modification is accomplished, ELP456 binds and uses ATP hydrolysis to release the modified tRNA substrate from ELP123[23] and complete the reaction (Fig. 6d). Despite the fact that all obtained structures of Elp3 proteins show an almost identical relative arrangement of the KAT and rSAM domain[13], we cannot exclude the possibility that the complex undergoes a short-lived conformation transition after acetyl-CoA hydrolysis.

Our work provides experimental insight into the molecular coordination of acetyl-CoA in the active site of ELP3 proteins. Previously, all evidence on acetyl-CoA coordination was exclusively based on experiments with the bacterial Elp3 homolog (DmcElp3), which acts as a homodimer and without the support of any other Elongator subunit[18,19,30]. Notably, a loop region that blocks the acetyl-CoA binding site in the apo structure of DmcElp3 had to be removed to achieve proper binding[30]. Here, we show that the analogous loop does not block the acetyl-CoA binding site in human ELP3. Instead of inhibiting acetyl-CoA binding, this loop provides crucial contacts to hold the adenosine 3′,5′-diphosphate moiety in place during the hydrolysis reaction. The tail of acetyl-CoA reaches deep into the catalytic cleft, where it positions the acetyl group close to Lys280. In both the generalized enzymatic and non-enzymatic mechanisms of lysine acetylation, a glutamic acid residue must attract protons from the reactive lysine before the acetyl transfer, followed by reprotonation of CoA by a tyrosine[47]. In the case of Elongator, however, the final target of the reaction is an RNA base, rather than a lysine.

We propose that the initial steps of acetyl-CoA hydrolysis in ELP3 follow the canonical mechanism, with Lys164 or Lys280 temporarily retrieving the acetyl group from acetyl-CoA, supported by our mass spectrometry data. Glu474 is highly conserved and could attract the proton from Lys280, but in principle $U_{33}$ could serve a similar purpose. Finally, the tyrosine pair Tyr529/Tyr530, as well as Tyr478 within the acetyl-CoA loop, would be perfectly positioned to reprotonate CoA after the hydrolysis reaction[47]. As we were unable to reconstitute the complete Elongator-mediated cm$^5$U$_{34}$ formation in vitro, we suspect that the concerted action of known accessory proteins [e.g., Kti11/Dph3[48], Kti12[49] and casein kinase 1[50]] are needed to accomplish the modification reaction. It is also possible that the phosphorylation status of ELP123 purified from insect cells might not support the final reaction and large-scale purifications from human cells might be needed in future studies.

tRNAs have a characteristic L-shape, and their modifying enzymes recognize specific tRNA species via highly diverse mechanisms[3]. Nonetheless, a protein-induced deformation of the modification site to flip out the target nucleotide seems to be a common feature among those enzymes[3]. Although $U_{34}$ is already exposed in its native U-turn conformation[51], ELP123 still deforms the ASL to flip out $U_{33}$. Strikingly, 97% of known tRNAs (except the initiator tRNA$^{fMet}$ and tRNA$^{Pro}_{GGG}$) contain an unmodified uridine at position 33[34]. The only reported exception is tRNA$^{Trp}_{CCA}$ in Leashmania tarentolae, where $U_{33}$ is thiolated at its second position[34]. Our data show that Elongator seems to exploit this unique tRNA identifier to trigger the initial stages of the modification reaction. Intriguingly, cm$^5$U and its derivatives (ncm$^5$U, mcm$^5$U, mchm$^5$U, ncm$^5$Um, mcm$^5$Um, and mcm$^5$s$^2$U) are only found in tRNAs and no other RNA families. Multiple contact points at the ASL, T-arm, and D-arm by ELP1/ELP3 in combination with the presence of a $U_{33}$/$U_{34}$ pair seem to be responsible for the high specificity of Elongator towards tRNAs. It remains to be shown whether other modifying

enzymes that target position 34 (i.e., CTU1/CTU2) or other ASL nucleotides employ $U_{33}$ in a similar mechanism.

Numerous pathogenic Elongator variants have been linked to severe human diseases, including cancer and neurodegenerative diseases[52]. Specific patient-derived mutations in ELP1[24,53], ELP2[54,55], ELP4[38,56], and ELP6[25,38] are causative for clinical phenotypes that can be recapitulated in mouse models. These mutations generally lead to reduced activity of Elongator in vitro and decreased tRNA modification levels in vivo. However, the observed phenotypes vary in severity and cover a broad spectrum of clinical manifestations, ranging from ataxia, microcephaly, and autism-like syndromes to severe intellectual disabilities. Furthermore, the two Elongator subcomplexes seem to have different importance for different neuronal cell types and differentiation stages[38]. Our work contributes an analysis of mutations in the enzymatically active ELP3 subunit, which are directly related to the integrity and activity of the Elongator complex in vitro. This suggests that the detected variants in ELP3 could indeed be disease-causing by altering the function of Elongator. Moreover, our high-resolution cryo-EM structure of human ELP123 will facilitate the functional assessment of newly detected variants in the future to support molecular diagnostics and studies using reverse genetics. Last but not least, ELP3 has been shown to be overexpressed in human cancer cells[57,58] and the function of Elongator promotes cancer cell survival during therapy relapse[59,60]. Thus, our high-resolution structures of the human Elongator complex importantly pave the way toward in silico screening for potential drugs in order to either ameliorate or inhibit Elongator activity in these diverse disease contexts.

## Methods

### Cloning and protein purifications

Codon-optimized ELP1 (O95163), ELP2 (Q61A86), and ELP3 (Q9H9T3) open reading frames were cloned into pFastBac1 HT A. The ELP3 clone coded for an in-frame Twin-Strep-Tag at the C-terminus of the ELP3 product. The ELP123 construct was generated using Gibson assembly by amplifying all three genes via PCR with primers adding specific overhangs on 5′ and 3′ sides of each expression cassette, allowing to determine the specific order of the ORFs. Subsequently, the amplified modules were assembled within the pBig1a plasmid. Mutations in ELP123 were introduced by QuikChange mutagenesis. The construct for the production of the ELP456 complex from Homo sapiens in insect cells was previously described with the similar procedure as mentioned above[38]. Genes encoding ELP4, ELP5, and ELP6 were cloned into pFastBac1 HTa plasmids with Flag-TEV-ProteinA tag sequence localized on the 3′ side of ELP6 gene. These ELP4, ELP5, and ELP6 genes were then PCR amplified using individual-specific primers adding overhands on 5′ and 3′ sides for subsequent assembly into pBig1b expression vector. For ELP123 and ELP456 protein expression, SuperSf9-3 cells, and Hi5 cells were infected with multiplicity of infection (MOI) = 1 and grown for 3 days at 27 °C on a shaking platform, respectively. Subsequently, insect cells were lysed in Lysis Buffer (for ELP123: 50 mM HEPES pH 7.5, 100 mM NaCl, 2 mM DTT, 5% glycerol, DNase I, protease inhibitors; for ELP456: 50 mM HEPES pH 7.5, 150 mM NaCl, 2 mM $MgCl_2$, 2 mM DTT, 5% glycerol, DNase I, protease inhibitors) by 3 cycles of freezing and thawing in liquid nitrogen and sonication, followed by centrifugation (4 °C; 80,000 × g). ELP123 variants were purified using StrepTrap HP 5 ml column (GE Healthcare) eluted in Strep Elution Buffer (50 mM HEPES, 100 mM NaCl, 1 mM DTT, 10 mM d-desthiobiotin, pH 7.5), followed by affinity chromatography on HiTrap Heparin HP 5 ml column (GE Healthcare) and elution in a gradient of Heparin Elution Buffer (50 mM HEPES, 1 M KCl, 1 mM DTT, pH 7.5). Finally, eluates were run on Superose 6 Increase 10/300 GL column (GE Healthcare) in 20 mM HEPES pH 7.5, 100 mM NaCl, 2 mM DTT. ELP456 supernatants were purified on IgG agarose beads (Merck) followed by 1 h Tobacco Etch Virus (TEV) protease cleavage in Cleavage Buffer (50 mM HEPES pH 7.5, 150 mM NaCl, 2 mM $MgCl_2$, 2 mM DTT). The protein sample was then

applied to a Superdex 200 Increase 10/300 GL column (GE Healthcare) equilibrated in 20 mM HEPES pH 7.5, 150 mM NaCl, 2 mM $MgCl_2$, 5 mM DTT. Selected fractions were pooled and concentrated with an Amicon Ultra-0.5 (100 kDa cut-off) concentrator. Aliquots were frozen in liquid nitrogen and stored at −80 °C for further use.

## tRNA production (in vitro transcription reaction)

The tRNA was produced using the T7 RNA polymerase-mediated run-off method[61]. The DNA template contained a T7-promoter sequence and followed by the tRNA sequences (tRNA$^{Gln}_{UUG}$: GGCCCCATGGTG-TAATGGTTAGCACTCTGGACTTTGAATCCAGCGATCCGAGTTCAAATC TCGGTGGGACCTCCA;

tRNA$^{Gln}_{CUG}$: GGTTCCATGGTGTAATGGTTAGCACTCTGGACTCT-GAATCCAGCGATCCGAGTTCAAATCTCGGTGGAACCTCCA;

tRNA$^{Ser}_{UGA}$: GCAGCGATGGCCGAGTGGTTAAGGCGTTGGACTTGA AATCCAATGGGGTCTCCCCGCGCAGGTTCGAACCCTGCTCGCTG CGCCA;

tRNA$^{Arg}_{UCU}$: GTCTCTGTGGCGCAATGGACGAGCGCGCTGGACTTC TAATCCAGAGGTTCCGGGTTCGAGTCCCGGCAGAGATGCCA;

tRNA$^{Glu}_{UUC}$: TCCCTGGTGGTCTAGTGGCTAGGATTCGGCGCTTTC ACCGCCGCGGCCCGGGTTCGATTCCCGGTCAGGGAACCA;

tRNA$^{Lys}_{UUU}$: GCCCGGATAGCTCAGTCGGTAGAGCATCAGACTTTT AATCTGAGGGTCCAGGGTTCAAGTCCCTGTTCGGGCGCCA).

The in vitro transcription reaction was performed in a 500 μL volume containing DNA template, T7 RNA polymerase, and reaction buffer (20 mM Tris, pH 8.0, 5 mM DTT, 150 mM NaCl, 8 mM $MgCl_2$, 2 mM spermidine, 20 mM NTPs, RNasin, and pyrophosphatase). The reaction was performed at 37 °C overnight and followed by DNase I treatment to remove DNA templates. The product was then purified using a DEAE column and heat treatment at 80 °C for 2 min and followed by the slow cooling process to room temperature as the re-annealing process. To obtain a homogenous tRNA population, the samples were subjected to a Superdex 75 Increase gel filtration column, and the tRNA-containing fractions were pooled and stored at −80 °C. For MST assays, the internally Cy5-labeled in vitro-transcribed human tRNAs were produced as mentioned above where the additional 5% of Cy5-CTP was introduced in the reaction.

## Microscale thermophoresis (MST)

The Cy5-labeled tRNA$^{Gln}_{UUG}$ (14 or 30 nM) was incubated with serial dilutions of purified *Min*Elp3 or ELP123 variants (starting from 1.5 μM) in MST Buffer (20 mM HEPES, 100 mM NaCl, 5 mM DTT, pH 7.5, 0.05% Tween 20) at 4 °C for 30 min[61]. The samples were applied to capillaries (MO-K025, Nanotemper Technologies) and the measurements were performed using Monolith Pico by MO.Control v2.5.4 (Nanotemper Technologies) with 60% excitation power at 25 °C. Obtained data were analyzed and dissociation constant values were calculated using MO.2 Affinity software v1.1 (Nanotemper Technologies) from at least three independent repeats. The graphs were prepared using Prism v8.0.2 (GraphPad) software.

## Acetyl-CoA hydrolysis assay

Purified ELP123 (0.475 μM) was mixed with 10 μM in vitro-transcribed tRNA in presence of 500 μM acetyl-CoA in 1× acetyl-CoA Assay Buffer (MAK039, Merck) and incubated in a thermocycler for 30 min at 37 °C[30]. To remove proteins and tRNAs, the samples were passed through a 3 kDa cut-off concentrator (EMD Millipore). The flow-through was collected and subjected to an acetyl-CoA assay kit (MAK039, Merck) for quantitation determinations. The reactions were performed according to the manufacturer's instructions. Fluorescence intensity was measured using a plate reader (TECAN) at the probe-specific excitation (535 nm) and emission (587 nm) wavelengths. The measurements for individual conditions were calculated from at least three independent experiments. The graphs were prepared using Prism v8.0.2 (GraphPad) software.

## Cryo-EM sample preparation and single-particle analyses

Cryo-EM samples were prepared using a similar method as described previously[27]. Briefly, freshly prepared proteins (0.6 g/l) without or with tRNAs (3.3 μM), acetyl-CoA/DCA/ECA (0.5 mM) and SAM (1 mM) were mixed in the binding buffer (20 mM HEPES, pH 7.5, 100 mM NaCl, 2 mM $MgCl_2$, 2 mM DTT) and applied to glow-discharged holey carbon grids (Quantifoil R2/1), which prior to that were glow-discharged (8 mA, 60 s). The vitrifications were carried out in Thermo Fisher Mark IV Vitrobot set to 4 °C and 100% humidity, with 15 s incubation time, blot force 5, and blot time 5 s. Grid screening and data collection were carried out at the SOLARIS National Synchrotron Radiation Centre UJ, Krakow, Poland. For the data collection, a Thermo Fisher Titan Krios G3i microscope (EPU v.2.10.0.1941REL) operating at an accelerating voltage of 300 kV equipped with Falcon III and K3 direct detectors was used to collect 5300, 4214, 20720 and 3192 micrographs of tRNA-free ELP123 and tRNA bound with acetyl-CoA, DCA and ECA, respectively. For each dataset, subsets of potential particles were picked from a small randomized subset of micrographs using CryoSPARC (v3.3.0) blob picker and used for TOPAZ training. The resulting TOPAZ models were then used to pick particles from the same subsets of the micrographs, and after particle curation, new models were trained - this process was repeated iteratively. To maximize the number of particles for each dataset, all TOPAZ models were used to pick particles from the full datasets. Next, 2D and 3D classifications in cryoSPARC were used to remove junk particles as well as duplicates from the initial picks. Elp123 particles were subclassified in CryoSPARC[62] to obtain the final 3D reconstructions. Map sharpening was performed in either RELION (v3.1) or using deepEMhancer software (v0.14). PDB models were generated either by SWISS-MODEL server using yeast and human existing structures, or AlphaFold2 predictions. These models were fitted into the density using Namdinator (v2.12) for flexible fitting[29,62]. Models were manually corrected and refined in COOT (v0.9.7 EL)[63], with final automated refinement in PHENIX (v1.19.2-4158)[64]. The structures were visualized using PyMOL (v1.7) or UCSF ChimeraX (v1.2.5)[65,66].

## Yeast genetic manipulations and phenotypic characterization

Genomic mutations at the *ELP3* locus in yeast strain UMY2893 (Table S1) were generated as previously described[23]. In short, site-directed mutagenesis was carried out on a plasmid carrying wild-type *ELP3* together with a *KlTRP1* marker, and the substitutions were introduced into the *elp3Δ::KlURA3* target locus. The correct insertion was confirmed via PCR and sequencing. Phenotypic characterization was carried out by the *GAL1* inducible γ-toxin plasmid[37] which encodes the tRNase subunit (γ-toxin) of zymocin. Serial cell dilutions were spotted on either 2% glucose or galactose-containing media and documented after 3 days of cultivation at 30 °C.

## tRNA isolation from yeast

100 ml cultures for each biological triplicate were grown in yeast extract-peptone-dextrose media to an optical density (OD$_{600\,nm}$) of 1, pelleted at 4000 × g for 2 min, washed and resuspended in 1.5 ml NucleoZOL (Macherey Nagel). Mechanical lysis was achieved by adding 500 μl glass beads (0.5 mm diameter) and bead beating for 6 cycles each 60 s. 20% v/v chloroform was added and bead beated for another 60 s. The suspension was incubated for 15 min on ice, centrifuged at 21,000 × g for 30 min and the aqueous phase was separated. To precipitate large RNA, 2/3 volume of 8 M LiCl was added and incubated at −20 °C for 3 h. After a centrifugation step at 21,000 × g for 30 min at 4 °C, the supernatant was separated and the LiCl precipitation step was repeated. The resulting tRNA-containing supernatant was precipitated by adding 10% v/v 3 M sodium acetate pH 5.2, 2.5 volumes 100% ethanol, and incubation overnight at −20 °C. tRNA was pelleted by centrifugation as before, air dried, and resuspended in 10 mM sodium acetate pH 5.2. tRNA concentrations were determined by using the Epoch (Agilent BioTek) spectrophotometer and subsequently stored at −80 °C.

### tRNA cleavage by γ-toxin

As previously described[23,67], GST-γ-toxin was used to cleave $mcm^5s^2U_{34}$ modified tRNAs and thus, assess the tRNA modification activity of Elongator in *elp3* mutants. tRNA levels were adjusted to 2 µg in toxin cleavage buffer [20 mM Tris-HCl (pH 8.0) at 4 °C, 150 mM NaCl, 2 mM EDTA and 2 mM DTT]. One half was separated as a loading control whereas the other was incubated with either 4 µM GST or GST-γ-toxin at 30 °C for 30 min. The RNA was denatured in loading buffer (20 mM Tris-HCl (pH 8.0) at 4 °C, 4 M urea, 20 mM boric acid, 2 mM EDTA, 0.02% w/v xylencyanol ff, 0.01% w/v bromophenol blue) at 90 °C for 5 min. Cleaved tRNA were separated from intact tRNA by 10% urea-PAGE (100 mM Tris-HCl (pH 8.0) at 4 °C, 100 mM boric acid, 1 mM EDTA, 10% acrylamide w/v (acrylamide/bis-acrylamide 19:1), 5.12 M urea) at 180 V for 35 min in TBE buffer (100 mM Tris-HCl (pH 8.0) at 4 °C, 100 mM boric acid, 2 mM EDTA). tRNA was visualized by SYBR™ Gold (Invitrogen) staining and documented on the FastGene blue/green LED transilluminator DE (Nippon Genetics).

### *SUP4* plasmid construction and functional analyses

The sequence of *SUP4*± 200 bp was retrieved from UMY2893[17] (Table S1) by PCR, agarose gel extraction (NIPPON Genetics), and subsequently treated with T4-polynucleotid kinase (Thermo Scientific). The backbones of single-copy YCplac111 and multicopy YEplac181[68] were linearized via PCR, DpnI digested (New England Biolabs), and purified from agarose gel (NIPPON Genetics). Both, insert and vector were blunt ligated by T4 DNA Ligase (Thermo Scientific), transformed in *E. coli,* and confirmed via sequencing. The U33C mutation in *SUP4* was installed via PCR and verified by sequencing.

Plasmids were transformed into yeast strain W303-1B (Table S2) to test for *SUP4* tRNA suppressor and *ochre* readthrough function from single- and multicopy plasmids. The phenotypic analyses were carried out as previously described[49]. Briefly, the UAA *ochre* stop codon readthrough is dependent on the $mcm^5U_{34}$ modification in *SUP4* which codes for a suppressor tRNA$^{Tyr}_{U\psi A}$ variant with a $G_{34}$ to $U_{34}$ exchange[17]. In the W303-1B background, translation of the *ade2-1* and *can1-100 ochre* (UAA) reporter alleles is prematurely terminated. The Elongator-dependent $mcm^5$ modification of *SUP4* allows for *ochre* readthrough and therefore enables respectively, adenine prototrophy and uptake of extracellular arginine. The former trait can be positively selected for on growth medium lacking adenine, the latter can be negatively selected against by supplementing the growth medium with the cytotoxic arginine analog canavanine.

### HPLC analyses of RNA nucleotides

W303-1B wild-type strain transformed with multicopy empty vector, tRNA$^{SUP4}$ wild-type or U33C mutant, and *Δelp3* transformed with tRNA$^{SUP4}$ wild-type, were grown in 1 L synthetic drop-out media and harvested at an $OD_{600\ nm}$ 1.5. Bulk yeast tRNA was extracted as described above. Individual tRNA$^{SUP4}$ was isolated from 600 µg bulk tRNA, using a biotinylated oligonucleotide specific for *Sc*tRNA$^{Tyr}$. Isolated tRNA$^{SUP4}$ (400 ng) was digested with 10 U Nuclease P1 (New England Biolabs) overnight at 37 °C in 30 mM ammonium acetate (pH 5.3) and 0.2 mM $ZnCl_2$, followed by digestion with 1 U bacterial alkaline phosphatase (Sigma−Aldrich) for 4 h at 37 °C. Conversion of $mcm^5U$ to $cm^5U$, was carried out as described[69]. Briefly, digested tRNA controls and $mcm^5U$ standards were incubated with 1 M NaOH for 8 min at room temperature, then neutralized with 1 M acetic acid and subjected to HPLC analysis. Nucleosides were analyzed by HPLC (Agilent 1200 HPLC system equipped with a diode array detector at 254 nm) using a C-30 reverse phase column (Develosil 5 µm RP-Aqueous C30 250 × 4.6 mm, CHO-5690, Phenomenex) at 20 °C. Mobile phase A consisted of 10 mM ammonium acetate pH 5.3 and 2.5% methanol (v/v) and mobile phase B was 10 mM ammonium acetate pH 5.3 and 20% methanol (v/v). The elution was performed at a flow rate of 1 ml/min using a linear gradient: 0−12 min 0% buffer B, 12−20 min 0% buffer B, 20−25 min 10% buffer B, 25−32 min 25% buffer B, 32−36 min 60% buffer B, 36−45 min 62% buffer B, 45−56 min 100% buffer B. The column was restored to initial conditions for 17 min with 100% buffer A.

### Photo-crosslinking mass spectrometry

85 µg of *Hs*Elongator at 1.4 µM concentration were crosslinked with sulfo-SDA (sulfosuccinimidyl 4,4′-azipentanoate, Thermo Scientific) at 1:500 and 1:1000 protein:crosslinker molar ratios. For *Mm*Elongator, 60 µg at 5.0 µM concentration were used. After incubation at room temperature for 60 min, the reaction mix was irradiated with UV light at 365 nm using a Luxigen34 LZ1 LED emitter (Osram Sylvania Inc.) for 10 s. 50 mM ammonium bicarbonate was added to quench the reaction. SDS-PAGE was used to separate the crosslinked complexes from single subunits (Novex Bis-Tris 4−12% SDS−PAGE gel, Life Technologies), and gel sections containing the crosslinked complex were excised. Dithiothreitol reduced the sample and iodoacetamide alkylated free sulfhydryl groups[70]. Proteins were digested using 1 µg trypsin (Thermo Scientific Pierce) per 20 µg of protein sample. The resulting peptides from the digestion were subjected to desalting using C18 StageTips[71].

The eluates from the StageTips were fractionated via size exclusion chromatography (SEC) using a Superdex™ 30 Increase 3.2/300 column (GE Healthcare). A mobile phase comprising 30% (v/v) acetonitrile (ACN) and 0.1% trifluoroacetic acid at a flow rate of 10 µl/min was used. The first six 50 µL fractions containing peptides were collected and the solvent was removed using a vacuum concentrator.

### Crosslinking mass spectrometry acquisition

As a liquid chromatography-tandem mass spectrometry (LC-MS/MS) system, an Ultimate 3000 RSLCnano system (Dionex, Thermo Fisher Scientific) connected to an Orbitrap Fusion Lumos Tribrid mass spectrometer (Thermo Fisher Scientific) was utilized. Each fraction obtained from SEC was resuspended in 3.2% (v/v) ACN and 0.1% (v/v) formic acid. The resuspended samples were then injected into a 50-cm EASY-Spray C18 LC column (Thermo Scientific) at an operating temperature of 50 °C. The flow rate for both sample loading and separation was set at 0.3 µl/min. The mobile phase consisted of solvent A (0.1% (v/v) formic acid) and solvent B (80% (v/v) ACN, 0.1% (v/v) formic acid). A gradient ranging from 2% to 55% B was applied over a period of 90−100 min, with specific optimization for each SEC fraction. Subsequently, the percentage of B was increased to 95% within 2.5 min. The eluting peptides were ionized using an EASY-Spray source (Thermo Scientific) before being introduced to the mass spectrometer. Two acquisitions were performed for each SEC fraction.

The data-dependent MS acquisition was conducted in cycles of 2.5 s, utilizing the top-speed setting. The full scan mass spectrum was recorded in the Orbitrap with a resolution of 120,000. Ions with charge states ranging from 3+ to 7+ were selected for fragmentation via stepped higher-energy collisional dissociation (26%, 28%, and 30%), employing a decision tree approach prioritizing charge states 4+ to 7+[72]. For the first injection, precursors were sorted by intensity, and for the second by highest charge. The resulting fragment spectra were recorded in the Orbitrap with a resolution of 60,000. To avoid redundancy, peaks were excluded after a single occurrence for a duration of 60 s (dynamic exclusion).

### Crosslinking mass spectrometry data processing

The mass spectrometric raw data were processed to generate MS2 peak lists using the MSConvert module of ProteoWizard (version 3.0.22194). Recalibration of precursor and fragment m/z values was performed based on the average mass error of linear peptide spectrum matches. The sequence databases for the crosslink identification and open modification searches were based on the 100 proteins with the highest intensity-based absolute quantification identified in the same acquisitions[73] (version 2.3.0.0). The sequences were cropped to the

mature polypeptide chain according to the 'Chain' feature as described on UniProt. FragPipe (version 18.0) in conjunction with MSFragger[74] (version 3.7) and philosopher[75] (version 4.8.1) were used to identify peptide modifications using the default settings of the 'Open' pre-set workflow. Modifications occurring in more than 2% of peptide spectrum matches were included in the crosslink identification search.

Crosslinked peptides were identified using xiSEARCH (version 1.7.6.4)[76]. The search parameters were set as follows: Enzyme: Trypsin; Missed Cleavages: 3; Missing Mono-Isotopic Peaks: 2; Cross-Linker: NonCovalent (0 u), SDA (82.04186484 u); MS Tolerance: 3 ppm; MS2 Tolerance: 5 ppm; Fixed Modifications: carbamidomethylation (C; 57.021464 u); Variable Modifications (* identified in open modification search): SDA-hydro* (K, S, T, Y; 100.05243 u), SDA-loop* (K, S, T, Y; 82.04186484 u), oxidation (M; 15.99491463 u), acetylation* (K, C, S, T, N-terminus; 42.010565 u).

The search results were filtered using xiFDR (version 2.1.5.6)[76,77] to a final FDR of 3% at the residue pair level. Prior to FDR estimation, spectral matches were filtered with the following settings: peptide1 unique matched conservative >2.0, peptide2 unique matched conservative >2.0, peptide1 unique crosslinked matched non lossy >0.0, and peptide2 unique crosslinked matched non lossy >0.0. The "boost" feature was enabled for heteromeric crosslinks. The dataset comprises 1011 residue pairs, of which 299 heteromeric for *Hs*Elongator, and 361, of which 99 heteromeric for *Mm*Elongator. To visualize the crosslinks we used the mouse Elongator model from our previous work[23], and a human Elongator model assembled from the ELP123–tRNA–acetyl-CoA structure and ELP456 generated with SWISS-MODEL, which were flexible fit into a 9 Å low passed mouse Elongator cryo-EM map using Namdinator. Both mouse and human models were further auto-refined in PHENIX.

## Statistics

Graphed datasets are expressed as mean ± standard error of the mean (SEM) from three independent experiments. Statistical analysis was performed with GraphPad Prism software (version 7.05) using a one-way ANOVA ($\alpha = 0.05$) with Tukey's test or Dunnett's multiple comparisons test. Statistically significant differences are indicated.

## Reporting summary

Further information on research design is available in the Nature Portfolio Reporting Summary linked to this article.

## Data availability

The data supporting the findings of this study are available from the corresponding authors upon request. The micrographs, atomic coordinates, and the cryo-EM map of the ELP123–tRNA$^{Gln}_{UUG}$–acetyl-CoA complex have been deposited at the Electron Microscopy Public Image Archive (EMPIAR-11650), the Protein Data Bank (PDB ID 8PTX) and the Electron Microscopy Data Bank (EMD-17924). The atomic coordinates and cryo-EM map of the ELP123, ELP123–tRNA$^{Gln}_{UUG}$–ECA and ELP123–tRNA$^{Gln}_{UUG}$–DCA have been deposited to the Protein Data Bank (PDB ID 8PTY, 8PTZ, 8PU0) and the Electron Microscopy Data Bank (EMD-17925, EMD-17926, EMD-17927), respectively. The crystal structure of tRNA (PDB 1EHZ) is available at https://www.rcsb.org/structure/1EHZ. The mass spectrometry data for ELP3 acetylation were deposited to the MassIVE repository with the dataset identifier MSV000092998 (https://doi.org/10.25345/C5N01044H). All cross-linking mass spectrometry data for mouse (ID JPST002342) and human (ID JPST002341) Elongator are available at (https://repository.jpostdb.org). Source data are provided with this paper.

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

## Acknowledgements

We thank Jakub Nowak and Lukasz Koziej for insightful discussions during the analyses. We thank Konrad Jazgar for the initial cloning assistance. We thank Roland Klassen for discussion and suggestions and Anders Byström for the kind gifts of yeast reporter strain UMY2893 and the tRNase (gamma-toxin) clone. In addition, we thank the MCB structural biology core facility (supported by the TEAM TECH CORE FACILITY/2017-4/6; S.G. grant from Foundation for Polish Science) for providing support and computational infrastructure. P.B. and D.S. were supported by PhD scholarships from respectively, Deutsche Forschungsgemeinschaft (DFG, Germany) and Otto Braun-Fonds (Germany). R.S. greatly acknowledges support by a DFG Elongator project (SCHA750/25-1) and receives funds as part of DFG consortium Biological Clocks on Multiple Time Scales (GRK2749-1). The work was supported by the European Research Council (ERC) under the European Union's Horizon 2020 research and innovation program (grant agreement No 101001394; S.G.). The work is supported under the Polish Ministry and Higher Education project: "Support for research and development with the use of research infrastructure of the National Synchrotron Radiation Centre SOLARIS" under contract nr 1/SOL/2021/2. The open-access publication of this article was funded by the Priority Research Area BioS under the program "Initiative of Excellence—Research University" at the Jagiellonian University in Krakow.

## Author contributions

Conceptualization: R.S., T.Y.L., S.G. Methodology: N.A., M.J., D.D., A.C.G., A.H., D.S., P.B., A.R., M.R., P.I., G.W., B.S.R., U.J., J.R. Investigation: N.A., M.J., D.S., P.B. Visualization: N.A., M.J., T.Y.L. Supervision: J.R., R.S., T.Y.L., S.G. Writing—original draft: N.A., M.J., T.Y.L., D.S., P.B. Writing—review & editing: N.A., M.J., R.S., T.Y.L., S.G.

## Competing interests

The authors declare no competing interests.
