## [Peer Review File · Nature Communications]

Cryo-EM structures of the human Elongator complex at workREVIEWER COMMENTS

Reviewer #1 (Remarks to the Author):

The paper submitted by Abassi and colleagues presents the first high resolution view of human Elongator, a acetyltransferase protein complex whose main function is to modify the wobble uridine of specific tRNAs. This 'tour de force' from the Glatt lab provides a series of cryo-EM structures of Elongator with tRNA and cofactors at different stages of the enzymatic reaction. Importantly, they also provide an in-depth analysis of various mutants, which they study from a structural, but also from a functional point of view. This important work allows, for the first time, to understand the different steps of the complex modification reaction and highlights the key mechanisms allowing the recognition of tRNA substrates, the hydrolysis of Acetyl-CoA and the atypical transfer of the acetyl group to the tRNA wobble uridine. Finally, they identify and study the impact of patient-derived mutations on the tRNA modification activity of Elongator.

The work is compelling, very well presented, accurate and the conclusions are correctly formulated. It is of great importance for the understanding of complex tRNA modification reactions, which biological importance is growing in many fields of research as well as in complex human diseases.

P. CLOSE

Reviewer #2 (Remarks to the Author):

In this manuscript the authors present a series of structures of the human Elongator in complex with tRNA and cofactors at different stages of the complex reaction cycle. The Elongator complex is large multicomponent system that modifies uridines at the wobble position (U34) in tRNAs to 5-carboxymethyluridine. This modification contributes to the accuracy of decoding and modification levels have been associated with several human diseases. The Elongator complex is composed of two copies of six different subunits (ELP1-6), which form two stable subcomplexes Elp123 and Elp456. The authors determined cryo-EM structures of human Elp123 on its own and bound to tRNA and acetyl CoA, revealing that the overall structure of Elp123 is well conserved across eukaryotes. The ACO bound structure revealed that the ACO binding pocket within the ELP3 is formed by the ACO-loop which is disordered in the apo structure. Next the authors determined additional structures with bound ECA and DCA ligands representing further steps in the ACO reaction. All three ligand bound structures are similar revealing that Elp3 does not undergo large-scale conformational changes during the reaction. The significance of several Elp3 residues was confirmed by tRNA binding and ACO hydrolysis assays. The cryo-EM structures also revealed how Elp3 and Elp1 mediate tRNA binding and distort the anticodon stem loop. Intriguingly the authors observed that U33 from the ASL makes direct contact the Kat domain, suggesting that it may be important for substrate identification. The authors then demonstrated that replacement of U33 with a C completely abolishes activity in vitro. They also demonstrated the significance of U33 in yeast using a reporter strain and by isolating tRNAs and analyzing the modification patterns by HPLC. The cryo-EM structures also led to a proposed mechanism of acetyl transfer through a series of well conserved lysine and tyrosine residues. Using mass-spec the authors were able to detect acetylation of several of these residues. Mutation of these residues abolishes activity in vitro and in vivo. Finally, the authors mapped disease associated variants onto the structures, produced the recombinant complex, and performed activity assays with the variants for which the complex remained stable. Overall, this is a very comprehensive body of work that reveals exciting new insight into the structure and mechanism of the human Elongator complex. I fully support publication in Nature Communications once the authors address the following issues.

Major Issues:

- Fig. 2 – I like how the authors show density for the bound ligands, however the authors should be careful with which side chain residues and bonds they are showing in each panel, based on the local

resolution. If there is no well-ordered density for the side chains, then they should not be shown in the figure. If it's not too messy I recommend segmenting the cryo-EM maps and showing the density for the ACO loop and indicated side chains in addition to the ligands. I'm also confused by the location of Lys164, as shown it looks like it is part of the ACO loop but the loop is residues 477-497?

- The authors must include the map-model FSC curves for each cryo-EM structure.
- The authors must show the density of the modeled 5'-dA ligand in Fig. S5.
- The entire section on the ELP123 and ELP456 interaction is confusing and speculative. The figure illustrating the crosslinks is hard to read and follow. Given that the complex has a higher stoichiometry with two copies of each protein this further complicates the interpretation of the crosslinks. I recommend that the authors either majorly revise this section of the manuscript or consider removing it entirely. I do not think this data is necessary to support the major conclusions of their manuscript.

Minor Issues:

- Fig. S5b – It is very hard to distinguish between the different shades of pink.
- ELP vs Elp? The authors use both throughout.
- Is ACO the standard abbreviation for acetyl-CoA? Why not AcCoA or ACoA? This abbreviation is confusing in Fig. 7, where the authors show ACO turning into Ac + CoA.
- Fig. 3 – I recommend using the one letter AA abbreviations in the insets to label the specific residues. It would also be helpful to label some of the tRNA bases, esp U33 and U34.
- Fig. 4 – It would be nice to show the interaction between His476 and U33 in this figure.
- What is the higher MW band that co-purifies with the complex and is enriched in many of the mutants?
- Methods (Line 582) – Please give the tRNA concentration used in terms of μM .
- Methods (Line 593) – Please elaborate on TOPAZ picking – did you perform TOPAZ training?

Reviewer #3 (Remarks to the Author):

Anticodon U34 is modified at the 5 position of the pyrimidine nucleobase by a range of different alkyl groups, facilitates non-canonical pairing which is important for translational fidelity so this tuning of tRNA function by modification is critical. Elongator makes the initial carboxymethyl modification and understanding how it recognizes its substrates and assembles a catalytic complex is a significant area of RNA biology.

The authors report cryoEM structures of the eukaryotic elongator complex ELP123 bound to tRNA at different stages in the reaction cycle. While the free enzyme is flexible the ES complex structures are ordered and high resolution 2.9 Å. A series of functional studies in the form of binding and in vitro reactions are analyzed for the native enzyme and a series of mutants. The structures show that the anticodon loop structure is distorted to present the U to be modified. Conserved residues involved in the reaction are tested as are the effects of pathogenic mutations on activity. The authors further test roles of active site residues and used in vivo yeast reporter assays to confirm in vivo effects. In the end the authors attempt to articulate a model of the catalytic cycle which gives a wholistic picture of specificity and catalysis. This is a very complete and authoritative study a landmark in the area of tRNA modifying enzymes.

The weaknesses here are a lack of kinetic detail to support the mechanistic model and interpretation

of mutants. Also, the authors present a series of different experiments and techniques but lack a clear overall mechanistic take home message. The issue that I believe the authors should respond to in revising their manuscript include the following.

1. The authors draw mechanistic conclusions regarding the effects of mutants based on a binding measurement and a rate measured under a single set of conditions which could be better described and defined. Given the large conformational change upon binding its likely tRNA dissociation is slow. Did the authors check to see if the binding reaction came to completion? If not then the apparent K_d may be biased toward k_{on} or k_{off} depending on their relative magnitudes.
2. Is the tRNA getting modified in these reaction? Or are the two cofactors just being hydrolyzed? If the modification reaction is not occurring, is it really accurate to say we are looking at a catalytic cycle?
3. It is not clear how the ordered binding mechanism of the reaction cycle was determined, how does it follow from the data? A point by point description in the discussion could be better connected to the cartoon shown in the figures. Best would be to use multiple turnover kinetics which can readily demonstrate ordered binding versus random binding mechanisms in multiple substrate reactions. The lack of kinetic detail is a significant limitation for the mechanistic conclusions that can be drawn. How would a random binding mechanism for tRNA, SOM and ACO affect the interpretation of the mutagenesis results versus an ordered mechanism?
4. What was the rational for the choice of substrate concentrations in kinetic assays? How will the interpretation of the kinetic results be different some or all of the substrates are at saturating concentrations?
5. The HPLC quantification of mcm5U in vivo is not very convincing, the occurrence of the small peak is coincidental with activity, but this does not seem like strong evidence. Is there some more direct or quantitative method? Also, the amount of modified tRNA is very small if this peak is real. Is this sufficient to result in the suppression effect that is observed?
6. It is not clear what the crosslinking analysis adds any new information. Interpreting crosslinking in terms of functional dynamics is problematic and the results from the paper do not appear to be integrated into the overall story which seems to be driving toward a model of the catalytic cycle. It seems like the authors expected human elongator would resemble yeast and mouse, which seems incremental unless there are larger issues that can be better communicated.
7. The authors propose a lysine relay mechanism to move the acyl group from coA to the nucleobase substrate. However, many residues are being modified apparently all over the protein. Is this correct? If so how does this square with the authors hypothesis of a specific relay pathway?
8. With respect to the author's hypothetical relay mechanism, is it not also possible that there is a conformational change after ACO hydrolysis that the authors simply have not observed. It does not have to be long lived to be functionally important. It seems like a simpler model especially in light of the fact that so many residues are getting modified across the protein, not just the ones on the pathway.

Point-by-point response

Abbassi et al. NCOMMS-23-54649

Reviewer #1 (Remarks to the Author):

The paper submitted by Abassi and colleagues presents the first high resolution view of human Elongator, a acetyltransferase protein complex whose main function is to modify the wobble uridine of specific tRNAs. This ‘tour de force’ from the Glatt lab provides a series of cryo-EM structures of Elongator with tRNA and cofactors at different stages of the enzymatic reaction. Importantly, they also provide an in-depth analysis of various mutants, which they study from a structural, but also from a functional point of view. This important work allows, for the first time, to understand the different steps of the complex modification reaction and highlights the key mechanisms allowing the recognition of tRNA substrates, the hydrolysis of Acetyl-CoA and the atypical transfer of the acetyl group to the tRNA wobble uridine. Finally, they identify and study the impact of patient-derived mutations on the tRNA modification activity of Elongator.

The work is compelling, very well presented, accurate and the conclusions are correctly formulated. It is of great importance for the understanding of complex tRNA modification reactions, which biological importance is growing in many fields of research as well as in complex human diseases.

P. CLOSE

Response: We highly appreciate the very positive response by the reviewer and the recognition of the impact of our work for the Elongator and tRNA modification field.

Reviewer #2 (Remarks to the Author):

In this manuscript the authors present a series of structures of the human Elongator in complex with tRNA and cofactors at different stages of the complex reaction cycle. The Elongator complex is large multicomponent system that modifies uridines at the wobble position (U34) in tRNAs to 5-carboxymethyluridine. This modification contributes to the accuracy of decoding and modification levels have been associated with several human diseases. The Elongator complex is composed of two copies of six different subunits (ELP1-6), which form two stable subcomplexes Elp123 and Elp456. The authors determined cryo-EM structures of human Elp123 on its own and bound to tRNA and acetyl CoA, revealing that the overall structure of Elp123 is well conserved across eukaryotes. The ACO bound structure revealed that the ACO binding pocket within the ELP3 is formed by the ACO-loop which is disordered in the apo structure. Next the authors determined additional structures with bound ECA and DCA ligands representing further steps in the ACO reaction. All three ligand bound structures are similar revealing that Elp3 does not undergo large-scale conformational changes during the

reaction. The significance of several Elp3 residues was confirmed by tRNA binding and ACO hydrolysis assays. The cryo-EM structures also revealed how Elp3 and Elp1 mediate tRNA binding and distort the anticodon stem loop. Intriguingly the authors observed that U33 from the ASL makes direct contact the Kat domain, suggesting that it may be important for substrate identification. The authors then demonstrated that replacement of U33 with a C completely abolishes activity in vitro. They also demonstrated the significance of U33 in yeast using a reporter strain and by isolating tRNAs and analyzing the modification patterns by HPLC. The cryo-EM structures also led to a proposed mechanism of acetyl transfer through a series of well conserved lysine and tyrosine residues. Using mass-spec the authors were able to detect acetylation of several of these residues. Mutation of these residues abolishes activity in vitro and in vivo. Finally, the authors mapped disease associated variants onto the structures, produced the recombinant complex, and performed activity assays with the variants for which the complex remained stable.

Overall, this is a very comprehensive body of work that reveals exciting new insight into the structure and mechanism of the human Elongator complex. I fully support publication in Nature Communications once the authors address the following issues.

Response: We highly appreciate that the reviewer fully supports the publication of our work and recognizes that our study reveals exciting new insight into the structure and mechanism of the human Elongator complex. We also thank the reviewer for the very constructive suggestions that did further improve our manuscript – detailed responses to all raised issues are listed below.

Major Issues

•Fig. 2 – I like how the authors show density for the bound ligands, however the authors should be careful with which side chain residues and bonds they are showing in each panel, based on the local resolution. If there is no well-ordered density for the side chains, then they should not be shown in the figure. If it's not too messy I recommend segmenting the cryo-EM maps and showing the density for the ACO loop and indicated side chains in addition to the ligands. I'm also confused by the location of Lys164, as shown it looks like it is part of the ACO loop but the loop is residues 477-497?

Response: We have updated Figure 2 and added densities for the relevant amino acid side chains. In addition, we have also moved the labelling of Lys164 in the figure panels – as we now also show the respective density of Lys164, it is more obvious that this residue is not part of the acetyl-CoA loop.

•The authors must include the map-model FSC curves for each cryo-EM structure.

Response: We have updated the FSC plots for each of the cryo-EM reconstructions in Supplementary Figures S1, S2, S3 and S4 – now also showing the respective map-model FSC curves. The figure legends have been updated, accordingly.

•The authors must show the density of the modeled 5'-dA ligand in Fig. S5.

Response: We now show the densities of the modeled 5'-dA ligands in the updated Supplementary Figure S5. The figure legend has been updated, accordingly.

•The entire section on the ELP123 and ELP456 interaction is confusing and speculative. The figure illustrating the crosslinks is hard to read and follow. Given that the complex has a higher stoichiometry with two copies of each protein this further complicates the interpretation of the crosslinks. I recommend that the authors either majorly revise this section of the manuscript or consider removing it entirely. I do not think this data is necessary to support the major conclusions of their manuscript.

Response: We acknowledge the reviewer's comment. However, we think that the UV-induced chemical crosslinking mass spectrometry analyses of the fully assembled human and mouse Elongator complexes are still interesting to expert readers in the field. Our results not only identify dynamic regions, but also provide additional restraints to generate a model of the fully assembled human Elongator complex. Moreover, this model, which is based on experimental data, might be used to interpret the impact of clinically relevant mutations that reside in the interaction region.

Nonetheless, we agree with this reviewer (and reviewer 3) that this part slightly distracts from the major findings. Therefore, we have shortened the text section and moved the results into the Supplementary information section (Supplementary Figure S10), accordingly. Of note, we have placed data describing the additional analyses of the clinical point mutants into a separate Supplementary figure (Figure S11).

Minor Issues:

•Fig. S5b – It is very hard to distinguish between the different shades of pink.

Response: We agree with the reviewer and have updated the color scheme in Supplementary Figure S5b.

•ELP vs Elp? The authors use both throughout.

Response: We appreciate the reviewer's comment and would like to clarify the used (commonly accepted) nomenclature. We use "Elp" when talking about Elongator or Elongator proteins from different organisms (e.g. human, mouse, yeast, archaea or bacteria) in general. We use capitalized "ELP" exclusively when referring to the human Elongator complex or human Elongator proteins. Following the comment, we have double checked and confirmed the consistency throughout the manuscript text.

•Is ACO the standard abbreviation for acetyl-CoA? Why not AcCoA or ACoA? This abbreviation is confusing in Fig. 7, where the authors show ACO turning into Ac + CoA.

Response: We apologize for the confusion and we have now replaced "ACO" with "acetyl-CoA" throughout the text and in all figure panels.

•Fig. 3 – I recommend using the one letter AA abbreviations in the insets to label the specific residues. It would also be helpful to label some of the tRNA bases, esp U33 and U34.

Response: We acknowledge the suggestion by the reviewer. However, we intentionally use the three letter code for amino acid residues (e.g. Lys164) to discriminate the residue from amino acid substitutions (for which we use the one letter codes; e.g. K164A). We do fully agree with

the suggestion to label certain tRNA bases - therefore, we have added additional labels for some of the nucleotides in Figure 3.

•Fig. 4 – It would be nice to show the interaction between His476 and U33 in this figure.

Response: We now show a close up on His476 and U₃₃ in Supplementary Figure 6c.

•What is the higher MW band that co-purifies with the complex and is enriched in many of the mutants?

Response: We have analyzed the band by mass spectrometry and identified it as the acetyl-CoA carboxylase 1 (Acc1/ACACA). This potential contaminant has been found in Elongator preparations from different expression hosts and we studied it in greater detail during our previous work (Jaciuk et al., 2023). In short, Acc1 is a ~500 kDa multifunctional homodimeric enzyme, conserved and present in most eukaryotic species, including yeast and *S. frugiperda*. We use insect-derived cell lines as expression host to produce human ELP123. We use similar purification strategies for the yeast, mouse and human Elp123 complex and this protein co-purified in different quantities in most preparations – however, it appears to represent an impurity rather than a specific binding partner. This is supported by the fact that throughout all our cryo-EM analyses we did not find any 2D/3D classes for Acc1/ACACA. Moreover, we have also conducted Co-IP experiments in yeast, which did not show any specific interactions between Acc1 and Elongator (Jaciuk et al., 2023). Last but not least, we did not detect any specific crosslinks between Acc1 and mouse or human Elongator in our crosslinking mass spectrometry analyses. We now label the band in Supplementary Figure 5c and 11, describe it in the figure legends and added a reference to our previous work.

•Methods(Line 582) – Please give the tRNA concentration used in terms of μ M.

Response: We are sorry for the confusion - the used concentration of tRNA in the experiment was 3.3 μ M. The information has been added in the respective method section.

•Methods (Line 593) – Please elaborate on TOPAZ picking – did you perform TOPAZ training?

Response: The reviewer is absolute right – we performed TOPAZ training and we have updated the technical information, accordingly. The section now reads as follows - *“For each dataset, subsets of potential particles were picked from a small randomized subset of micrographs using CryoSPARC blob picker and used for TOPAZ training. The resulting TOPAZ models were then used to pick particles from the same subsets of the micrographs, and after particle curation, new models were trained - this process was repeated iteratively. To maximize the number of particles for each dataset, all TOPAZ models were used to pick particles from the full datasets. Next, 2D and 3D classifications in cryoSPARC were used to remove junk particles as well as duplicates from the initial picks.”*

Reviewer #3 (Remarks to the Author):

Anticodon U34 is modified at the 5 position of the pyrimidine nucleobase by a range of different alkyl groups, facilitates non-canonical pairing which is important for translational fidelity so this tuning of tRNA function by modification is critical. Elongator makes the initial carboxymethyl modification and understanding how it recognizes its substrates and assembles a catalytic complex is a significant area of RNA biology.

The authors report cryoEM structures of the eukaryotic elongator complex ELP123 bound to tRNA at different stages in the reaction cycle. While the free enzyme is flexible the ES complex structures are ordered and high resolution 2.9 Å. A series of functional studies in the form of binding and *in vitro* reactions are analyzed for the native enzyme and a series of mutants. The structures show that the anticodon loop structure is distorted to present the U to be modified. Conserved residues involved in the reaction are tested as are the effects of pathogenic mutations on activity. The authors further test roles of active site residues and used *in vivo* yeast reporter assays to confirm *in vivo* effects. In the end the authors attempt to articulate a model of the catalytic cycle which gives a wholistic picture of specificity and catalysis. This is a very complete and authoritative study a landmark in the area of tRNA modifying enzymes.

The weaknesses here are a lack of kinetic detail to support the mechanistic model and interpretation of mutants. Also, the authors present a series of different experiments and techniques but lack a clear overall mechanistic take home message. The issue that I believe the authors should respond to in revising their manuscript include the following.

Response: We highly appreciate that the reviewer has recognized that our work is a very complete and authoritative study in the area of tRNA modifying enzymes. We fully agree with the reviewer that a full kinetic characterization of the underlying modification reaction should and will be the ultimate scientific goal. However, we would like to point out that our molecular characterizations are still hampered by the fact that nobody has been able to reconstitute the U₃₄ modification reaction of the eukaryotic Elongator complex *in vitro*. Hence, we can only evaluate the initial steps of the reaction *in vitro*.

Without understanding the whole system, it is likely that we would wrongly interpret kinetics from individual steps, which are likely to differ in conditions that can promote an efficient reconstitution of the modification reaction. However, we strongly believe that our presented structural and molecular framework for substrate and ligand binding will enable us and other groups to explore the specific dynamic parameters of the reaction in future studies.

Of note, we also wished to easily (and with less expensive means) produce large quantities of the complex. We are still exploring all possibilities to reconstitute the reaction *in vitro* and we will follow this until we finally manage. However, this modification cascade is very complex and technically extremely challenging, including not only the Elongator complex itself, but also numerous regulatory proteins, co-factors, ligands and substrates. We hope the reviewer appreciates our tedious efforts to make progress and we are very optimistic that our current work is essential to fully understand the system and its regulation in the future.

We of course very much appreciate the comments of the reviewer and aim to provide detailed explanations and technical justification for the individually raised points in the responses below.

1. The authors draw mechanistic conclusions regarding the effects of mutants based on a binding measurement and a rate measured under a single set of conditions which could be better described and defined. Given the large conformational change upon binding its likely tRNA dissociation is slow. Did the authors check to see if the binding reaction came to completion? If not then the apparent K_d may be biased toward k_{on} or k_{off} depending on their relative magnitudes.

Response: We fully agree with the reviewer that the experimental setup for binding equilibrium needs to be carefully designed (Jarmoskaite et al., 2020). We indeed have tested the incubation duration for reaching the binding equilibrium by varying time (15/30/60 min) and temperature (4 °C or 37 °C). During these initial tests, all binding curves reached the same saturation curve - except for the reaction incubated for 60 min at 37 °C. We detected protein degradation and precipitation of purified human ELP123 complex after incubation for 20 minutes at 37 °C, which explains the issues. Therefore, we decided to set the condition for binding at 4 °C for 30 min, which allows for complex formation and to reach an equilibrium state. As indicated in the review (Jarmoskaite et al., 2020), the equilibration time for K_d falling in the nanomolar range typically takes place within seconds. As one may still argue that macromolecules often require additional conformational rearrangements for binding to the target and it takes longer to reach equilibrium, we used a 30 minute incubation time.

Indeed, we have observed that tRNA binds to ELP123 complex tightly. This is related to k_{on} and k_{off} values that determine the K_d . On the one hand, the k_{on} rate constant is difficult to measure, but it should have an upper limit due to the collision rate in solution, which is $1 \times 10^9 \text{ M}^{-1}\text{s}^{-1}$ (Corzo 2006). On the other hand, k_{off} is directly related to the half-life of the complex which is highly relevant to biological process though it is difficult to measure due to the time scale falls in millisecond (Tiwary et al., 2015). We are afraid that methods to reliably measure k_{off} value experimentally require relatively large quantities. It is beyond our current technical abilities to obtain the k_{off} for ELP123 as we are strained by obtaining sufficient amount of proteins. Furthermore, we would like to highlight that the ELP456 complex seems to play a role in releasing tRNA from ELP123 (Glatt et al., 2012; Jaciuk et al., 2023) and additional factors might also contribute to the release of modified tRNA *in vivo*. For instance, Kti12 (Krutyholowa et al., 2019), casein kinase 1 (Hrr25/Kti14; Landrock et al. in preparation) (Abdel-Fattah et al., 2015) or type 2A-phosphatase (Sit4) (Abdel-Fattah et al., 2015; Mehlgarten et al., 2009) may influence the association and dissociation rates. For all of the above reasons, we limited ourselves to analyzing the initial binding of different tRNA to ELP123 in presence of acetyl-CoA derivatives.

In short, we have carefully tested and optimized the binding conditions and we are confident that the analyses were performed after equilibration of the system. In full agreement with the reviewer, we have used the optimized condition for testing and comparing the activity of all mutants.

2. Is the tRNA getting modified in these reaction? Or are the two cofactors just being hydrolyzed? If the modification reaction is not occurring, is it really accurate to say we are looking at a catalytic cycle?

Response: We have extensively tried to reconstitute the reaction by supplementing the cofactors (i.e., SAM and acetyl-CoA) as well as tRNA and other accessory proteins (i.e., human DPH3 and CBR1) in our *in vitro* system. However, we could only observe acetyl-CoA hydrolysis and SAM cleavage, but we still could not detect the appearance of cm^5U_{34} in the tested conditions. This has been mentioned in our initial submission on page 18 - “*As we were unable to reconstitute the complete Elongator-mediated cm^5U_{34} formation in vitro, we suspect that the concerted action of known accessory proteins [e.g., Kti11/Dph3, Kti12 and casein kinase I] are needed to accomplish the modification reaction. It is also possible that the phosphorylation status of HsELP123 purified from insect cells might not support the final reaction and large-scale purifications from human cells might be needed in future studies.*” We appreciate the reviewer’s comment on the wording “catalytic cycle” and we have rephrased it on several occasions.

3. It is not clear how the ordered binding mechanism of the reaction cycle was determined, how does it follow from the data? A point by point description in the discussion could be better connected to the cartoon shown in the figures. Best would be to use multiple turnover kinetics which can readily demonstrate ordered binding versus random binding mechanisms in multiple substrate reactions. The lack of kinetic detail is a significant limitation for the mechanistic conclusions that can be drawn. How would a random binding mechanism for tRNA, SAM and ACO affect the interpretation of the mutagenesis results versus an ordered mechanism?

Response: We would like to highlight that the proposed reaction scheme is based on both, data from the presented manuscript and previously published work.

ELP123 was initially described 25 years ago (Otero et al., 1999) but the biochemical and biophysical characterizations were hampered due to the inability to obtain the recombinantly expressed proteins. Since then, most investigations have been carried out using homologs of Elp3 from bacteria (Glatt et al., 2016) or archaea (Lin et al., 2019; Paraskevopoulou et al., 2006; Selvadurai et al., 2014). From these studies of Elp3 homologues, we have learned a lot about the catalytic Elp3 subunit:

- 1) Elp3 can bind tRNAs directly
- 2) SAM-binding and SAM-cleavage requires the presence of an iron-sulfur cluster in the SAM domain of Elp3
- 3) Elp3 can bind to acetyl-CoA (approx. $K_d \sim 150 \mu M$), yet we speculate that the binding might be facilitated by the pre-requisite binding of tRNA to make the binding site accessible
- 4) Elp3 shows higher acetyl-CoA hydrolysis activity upon tRNA binding, which is likely due to the allosteric effect for acetyl-CoA binding

In summary, SAM, acetyl-CoA and tRNA can bind to Elp3 without the prerequisite of one another. However, acetyl-CoA hydrolysis is induced by the presence of tRNA, which is caused by local structural rearrangements of the active site that facilitate acetyl-CoA binding and turnover. As expected, we confirmed that acetyl-CoA hydrolysis can be induced by the

presence of tRNA in the human ELP123 complex system. By using two acetyl-CoA analogs (i.e., ECA: non-hydrolysable acetyl-CoA analog; DCA: mimicry of hydrolyzed acetyl-CoA), we propose a mechanism that is based on snapshots of the catalytic states including the initial state (ELP123–SAM), substrate-bound state (ELP123–tRNA–acetyl-CoA/ECA) and the post-modification reaction (ELP123–tRNA–DCA). Although we do not have an *in vitro* product formation assay in place, we have a well-defined yeast reporter assay with which we are able to monitor the cm⁵ modification status *in vivo*. This also serves as a cross-reference for our observations in the *in vitro* mutagenesis approach (see response to reviewer 3 point 5).

Nonetheless, we have rephrased the mechanistic description as suggested using “reaction” instead of “catalytic cycle” and highlight that we can only focus on the early stages of the reaction.

4. What was the rationale for the choice of substrate concentrations in kinetic assays? How will the interpretation of the kinetic results be different some or all of the substrates are at saturating concentrations?

Response: The concentration of the constant limiting component (e.g. ELP123) should be below the K_d to avoid artificial measurements (Jarmoskaite et al., 2020). As we are aware of this limitation, we have performed initial tests and rounds of optimization by varying the ELP123 concentration (e.g. 14 nM, 30 nM or 150 nM) for the MST binding assay. The MST assay format is a highly sensitive method due to the fluorescence detection technique, which allows the K_d determination even at pM concentration ranges. For all tested ELP123 concentrations, we obtained the same K_d value, indicating that our experimental setup is robust and can be used to determine approximate K_d values. We used 14 nM of ELP123 for all experiments to use the precious purified material most efficiently.

In the tRNA-triggered acetyl-CoA hydrolysis assay, the enzyme (i.e., ELP123) is the constant limiting component, while acetyl-CoA and tRNAs are “unlimited” substrates to saturate the reaction. We have determined the approximate K_d for the interaction between ELP123 and tRNA is around ~50 nM, while the K_d of acetyl-CoA (determined by ITC) for Elp3 is ~100 μ M (Lin et al., 2019). Due to the limited quantities of purified ELP123, we cannot measure acetyl-CoA binding to ELP123 by ITC. However, we assume that the K_d is in a similar range as observed before. Moreover, we speculate that tRNA may have an allosteric effect on the acetyl-CoA binding due to local structural changes (Lin et al., 2019), therefore the $K_{d, app}$ for acetyl-CoA in the presence of tRNA should be smaller (< 100 μ M). Based on the mentioned criteria, we setup the acetyl-CoA hydrolysis assay at concentrations of at least 5 times higher than the K_d : tRNA (10 μ M) and acetyl-CoA (500 μ M). Similar to the binding assay, we initially set the ELP123 concentration as low as 150 nM, however, we were not able to detect any meaningful changes in acetyl-CoA hydrolysis as we suspected that the readout of the assay is not sensitive enough and prone to have a large error. Therefore, we decided to increase the ELP123 concentration to 475 nM (due to the limited quantities, this is the highest concentration we could reach). In this range we could monitor the ELP123-mediated acetyl-CoA hydrolysis to consume almost 90% of given acetyl-CoA within 30 minutes incubation at 37 °C. Therefore, we used this setup for investigating and comparing the activity of ELP123 wild-type as well as mutants.

5. The HPLC quantification of mcm⁵U *in vivo* is not very convincing, the occurrence of the small peak is coincidental with activity, but this does not seem like strong evidence. Is there some more direct or quantitative method? Also, the amount of modified tRNA is very small if this peak is real. Is this sufficient to result in the suppression effect that is observed?

Response: HPLC measurements of mcm⁵U (and derivatives) have been the preferred method of detecting and confirming the presence of this type of modification *in vivo* – in our opinion, this is still the gold standard detection method in the field (Chen et al., 2011; Han et al., 2015; Huang et al., 2005). Of note, we have purified the *SUP4* tRNA to dissect the specific tRNA nucleotide composition. It is worth noting that the *SUP4* tRNA is 75 nucleotides long and contains a total of 18 uridines. Thus, when digested and analyzed by HPLC, one would expect a relatively small mcm⁵U peak.

Before the analyses, we have calibrated our HPLC runs using synthetic standards. In addition, we used alkaline hydrolysis to convert mcm⁵U into cm⁵U. In short, we used several approaches to independently validate the observed peaks in the HPLC profiles (**Figure 4e** and **Figure S7**):

1) We used mass standards for the four main nucleosides and the modified derivatives of uridine (i.e., cm⁵U and mcm⁵U) to determine the retention time of each nucleoside using identical run conditions.

2) To further validate the presence of mcm⁵U₃₄, we used saponification. Alkaline hydrolysis removes the methyl ester bond from the 5-methoxycarbonylmethyl group, resulting in the formation of 5-carboxymethyl, which can be detected by the appearance of the cm⁵U peak (~10 min retention time) and simultaneous disappearance of the mcm⁵U (~37 min retention time) signal.

3) We also have created yeast mutants (i.e., the *elp3* deletion strain and the tRNA mutant which cannot be modified) that do not generate or accept mcm⁵U on the target tRNA. Hence we do not observe any signals corresponding to mcm⁵U from these internal negative controls.

Considering the low abundance of mcm⁵U in tRNAs, there is a commonly used *in vivo* reporting assay to cross validate the existence of mcm⁵U from *in vitro* condition (Björk et al., 2007; Guy et al., 2014; Klassen & Schaffrath, 2018), which we also have implemented in our study (**Figure 4d**). In detail, the mcm⁵U modification on *SUP4* is crucial for cell growth in the designed conditions (*ade*^{2-1 ochre} and *can*^{1-100 ochre} selection). Through investigating the cell growth phenotype changes, our results provide the evidence that mcm⁵U formation indeed is related to Elongator functionality. We have updated the figure presentation to clarify the mcm⁵U peak (**Figure S7**).

6. It is not clear what the crosslinking analysis adds any new information. Interpreting crosslinking in terms of functional dynamics is problematic and the results from the paper do not appear to be integrated into the overall story which seems to be driving toward a model of the catalytic cycle. It seems like the authors expected human elongator would resemble yeast and mouse, which seems incremental unless there are larger issues that can be better communicated.

Response: Please see response to reviewer 2 point 4. We would like to keep the crosslinking data as it demonstrates the conserved complex assembly and explains the ELP456 coordination with ELP123 and its role in releasing tRNA, which is part of the catalytic mechanism. Moreover, we believe that the overall complex structure should allow readers to map the residues of interest. The data is now moved into Supplementary Figure 10.

7. The authors propose a lysine relay mechanism to move the acyl group from coA to the nucleobase substrate. However, many residues are being modified apparently all over the protein. Is this correct? If so how does this square with the authors hypothesis of a specific relay pathway?

Response: It is correct that we indeed observed several lysine residues being acetylated. We would like to stress that these detected modified lysine residues are located at the exterior of ELP3. Moreover, we believe that these modifications have taken place inside the insect cells, because the complex was directly sent for mass spectrometry identification after purification. However, it is unclear which acetyltransferase is responsible for the posttranslational modifications and whether they play any functional role. In contrast, Lys280 and Lys316 are buried in the catalytic pocket, and it is unlikely to be accessible for acetyltransferases. As the two lysine residues locate in the potential “acetyl group transfer path”, we proposed a lysine relay mechanism which remains speculative at this point and will require further investigations. We originally have specified the relay mechanism in the manuscript at page 14 – *“This notion indicates possible alternative routes for the acetyl-transfer, pointing to a complex network within the catalytic cleft that is ultimately necessary for the modification reaction.”*

8. With respect to the author’s hypothetical relay mechanism, is it not also possible that there is a conformational change after ACO hydrolysis that the authors simply have not observed. It does not have to be long lived to be functionally important. It seems like a simpler model especially in light of the fact that so many residues are getting modified across the protein, not just the ones on the pathway.

Response: A few years ago, our initial assumptions were indeed based on a simple movement/conformation change that brings the KAT and the rSAM domains of Elp3 in proximity. However, the structures of Elp3 proteins from bacteria, archaea, yeast, mouse and human, that we have determined so far in isolation by X-ray crystallography or in complex with other subunits of the complex by single particle cryo-EM all show almost identical arrangement of the two domains. Our work demonstrates that this conformation of Elp3 is able to bind tRNAs and promote the positioning of the modified RNA base close to the active site. We have simply never obtained any indication for an alternative conformation. However, we in principle agree with the reviewer that there could be a short-lived transient state which is difficult to trap in mechanic structural studies. Therefore, we added the following statement – *“Despite the fact that all obtained structures of Elp3 proteins show an almost identical relative arrangement of the KAT and rSAM domain13, we cannot exclude the possibility that the complex undergoes an extremely short-lived conformation transition after acetyl-CoA hydrolysis.”*

References

- Abdel-Fattah, W., Jablonowski, D., Di Santo, R., Thüring, K. L., Scheidt, V., Hammermeister, A., ten Have, S., Helm, M., Schaffrath, R., & Stark, M. J. R. (2015). Phosphorylation of Elp1 by Hrr25 Is Required for Elongator-Dependent tRNA Modification in Yeast. *PLoS Genetics*, *11*(1). <https://doi.org/10.1371/journal.pgen.1004931>
- Björk, G. R., Huang, B., Persson, O. P., & Byström, A. S. (2007). A conserved modified wobble nucleoside (mcm5s2U) in lysyl-tRNA is required for viability in yeast. *RNA*, *13*(8), 1245–1255. <https://doi.org/10.1261/rna.558707>
- Chen, C., Huang, B., Anderson, J. T., & Byström, A. S. (2011). Unexpected accumulation of ncm5u and ncm5s2u in a trm9 mutant suggests an additional step in the synthesis of mcm5u and mcm5s2u. *PLoS ONE*, *6*(6), 1–10. <https://doi.org/10.1371/journal.pone.0020783>
- Glatt, S., Létoquart, J., Faux, C., Taylor, N. M. I., Séraphin, B., & Müller, C. W. (2012). The Elongator subcomplex Elp456 is a hexameric RecA-like ATPase. *Nature Structural and Molecular Biology*, *19*(3), 314–320. <https://doi.org/10.1038/nsmb.2234>
- Glatt, S., Zabel, R., Kolaj-Robin, O., Onuma, O. F., Baudin, F., Graziadei, A., Taverniti, V., Lin, T. Y., Baymann, F., Séraphin, B., Breunig, K. D., & Müller, C. W. (2016). Structural basis for tRNA modification by Elp3 from *Dehalococcoides mccartyi*. *Nature Structural and Molecular Biology*, *23*(9), 794–802. <https://doi.org/10.1038/nsmb.3265>
- Guy, M. P., Young, D. L., Payea, M. J., Zhang, X., Kon, Y., Dean, K. M., Grayhack, E. J., Mathews, D. H., Fields, S., & Phizicky, E. M. (2014). Identification of the determinants of tRNA function and susceptibility to rapid tRNA decay by high-throughput in vivo analysis. *Genes and Development*, *28*(15), 1721–1732. <https://doi.org/10.1101/gad.245936.114>
- Han, L., Kon, Y., & Phizicky, E. M. (2015). Functional importance of Ψ38 and Ψ39 in distinct tRNAs, amplified for tRNA^{Gln}(UUG) by unexpected temperature sensitivity of the s2U modification in yeast. *RNA*, *21*(2), 188–201. <https://doi.org/10.1261/rna.048173.114>
- Huang, B., Johansson, M. J. O. O., & Byström, A. S. (2005). An early step in wobble uridine tRNA modification requires the Elongator complex. *Rna*, *11*(4), 424–436. <https://doi.org/10.1261/rna.7247705>
- Jablonowski, D., Hammermeister, A., ten Have, S., Scheidt, V., Helm, M., Thüring, K. L., Abdel-Fattah, W., Stark, M. J. R., Di Santo, R., & Schaffrath, R. (2015). Phosphorylation of Elp1 by Hrr25 Is Required for Elongator-Dependent tRNA Modification in Yeast. *PLoS Genetics*, *11*(1), e1004931. <https://doi.org/10.1371/journal.pgen.1004931>
- Jaciuk, M., Scherf, D., Kaszuba, K., Gaik, M., Rau, A., Kościelniak, A., Krutyhołowa, R., Rawski, M., Indyka, P., Graziadei, A., Chramiec-Głabik, A., Biela, A., Dobosz, D., Lin, T.-Y., Abbassi, N.-E.-H., Hammermeister, A., Rappsilber, J., Kosinski, J., Schaffrath, R., & Glatt, S. (2023). Cryo-EM structure of the fully assembled Elongator complex. *Nucleic Acids Research*, *51*(5), 2011–2032. <https://doi.org/10.1093/nar/gkac1232>

- Jarmoskaite, I., Alsadhan, I., Vaidyanathan, P. P., & Herschlag, D. (2020). How to measure and evaluate binding affinities. *ELife*, *9*, 1–34. <https://doi.org/10.7554/ELIFE.57264>
- Klassen, R., & Schaffrath, R. (2018). Collaboration of tRNA modifications and elongation factor eEF1A in decoding and nonsense suppression. *Scientific Reports*, *8*(1). <https://doi.org/10.1038/s41598-018-31158-2>
- Krutyhołowa, R., Hammermeister, A., Zabel, R., Abdel-Fattah, W., Reinhardt-Tews, A., Helm, M., Stark, M. J. R., Breunig, K. D., Schaffrath, R., & Glatt, S. (2019). Kti12, a PSTK-like tRNA dependent ATPase essential for tRNA modification by Elongator. *Nucleic Acids Research*, *47*(9), 4814–4830. <https://doi.org/10.1093/nar/gkz190>
- Lin, T.-Y., Abbassi, N. E. H., Zakrzewski, K., Chramiec-Głabik, A., Jemioła-Rzemińska, M., Różycki, J., & Glatt, S. (2019). The Elongator subunit Elp3 is a non-canonical tRNA acetyltransferase. *Nature Communications*, *10*(1), 1–12. <https://doi.org/10.1038/s41467-019-08579-2>
- Mehlgarten, C., Jablonowski, D., Breunig, K. D., Stark, M. J. R., & Schaffrath, R. (2009). Elongator function depends on antagonistic regulation by casein kinase Hrr25 and protein phosphatase Sit4. *Molecular Microbiology*, *73*(5), 869–881. <https://doi.org/10.1111/j.1365-2958.2009.06811.x>
- Otero, G., Fellows, J., Li, Y., de Bizemont, T., Dirac, A. M. G. G., Gustafsson, C. M., Erdjument-Bromage, H., Tempst, P., Svejstrup, J. Q., Yang, L., de Bizemont, T., Dirac, A. M. G. G., Gustafsson, C. M., Erdjument-Bromage, H., Tempst, P., Svejstrup, J. Q., Li, Y., de Bizemont, T., Dirac, A. M. G. G., ... Svejstrup, J. Q. (1999). Elongator, a multisubunit component of a novel RNA polymerase II holoenzyme for transcriptional elongation. *Mol. Cell*, *3*(1), 109–118. [https://doi.org/10.1016/S1097-2765\(00\)80179-3](https://doi.org/10.1016/S1097-2765(00)80179-3)
- Paraskevopoulou, C., Fairhurst, S. A., Lowe, D. J., Brick, P., Onesti, S., Fairhurst, S. A., Lowe, D. J., Brick, P., & Onesti, S. (2006). The Elongator subunit Elp3 contains a Fe4S4 cluster and binds S-adenosylmethionine. *Molecular Microbiology*, *59*(3), 795–806. <https://doi.org/10.1111/J.1365-2958.2005.04989.X>
- Selvadurai, K., Wang, P., Seimetz, J., & Huang, R. H. (2014). Archaeal Elp3 catalyzes tRNA wobble uridine modification at C5 via a radical mechanism. *Nature Chemical Biology*, *10*(10), 810–812. <https://doi.org/10.1038/nchembio.1610>
- Tiwary, P., Limongelli, V., Salvalaglio, M., & Parrinello, M. (2015). Kinetics of protein-ligand unbinding: Predicting pathways, rates, and rate-limiting steps. *Proceedings of the National Academy of Sciences of the United States of America*, *112*(5), E386–E391. <https://doi.org/10.1073/pnas.1424461112>

REVIEWERS' COMMENTS

Reviewer #2 (Remarks to the Author):

The authors have done an excellent job addressing my previous concerns. I fully support publication of this manuscript and I congratulate the authors on their impressive tour-de-force study on the human elongator complex! The cryo-EM structures, biochemistry, and mass-spec analysis provide significant new mechanistic details of this complex enzymatic reaction.

Minor Issue:

Fig S10 – It's very hard to read the protein names in the top panel so I recommend making this part of the figure bigger.

Reviewer #3 (Remarks to the Author):

The authors made significant effort to address the concerns raised regarding the rigor of the kinetics and binding experiments in their responses but few changes to the actual manuscript, which would have been an improvement.

Point-by-point response

Abbassi et al. NCOMMS-23-54649A

Reviewer #2 (Remarks to the Author):

The authors have done an excellent job addressing my previous concerns. I fully support publication of this manuscript and I congratulate the authors on their impressive tour-de-force study on the human elongator complex! The cryo-EM structures, biochemistry, and mass-spec analysis provide significant new mechanistic details of this complex enzymatic reaction.

Response: We thank the reviewer for recognizing our efforts to address the raised concerns and we are more than happy that the reviewer now fully supports publication of our manuscript.

Minor Issue:

Fig S10 – It's very hard to read the protein names in the top panel so I recommend making this part of the figure bigger.

Response: We thank the reviewer for spotting this issue – we fully agree and have increased the font size and the position of the labels in the upper panel of Fig S10.

Reviewer #3 (Remarks to the Author):

The authors made significant effort to address the concerns raised regarding the rigor of the kinetics and binding experiments in their responses but few changes to the actual manuscript, which would have been an improvement.

Response: We thank the reviewer for acknowledging our efforts and we hope to provide detailed kinetics of the underlying modification reaction in our future work.

Additional changes highlighted in the revised manuscript (editorial remarks)

Page 21 – we provided additional technical information

Pages 22/23 – we added the respective sequences of all used tRNAs

Pages 23/24/25 – we added the specific versions of used software

Page 36 – we added initials to the respective grants

Page 37 – we changed the layout/format of Table 1

Pages 38/39 – we revised the figure legends and provided additional information and abbreviations.